# AGENT-E: FROM AUTONOMOUS WEB NAVIGATION TO FOUNDATIONAL DESIGN PRINCIPLES IN AGENTIC SYSTEMS

## ABSTRACT

Web agents that can automate complex and monotonous tasks are becoming essential in streamlining workflows. Due to the difficulty of long-horizon planning, abundant state spaces in websites, and their cryptic observation space (i.e. DOMs), current web agents are still far from human-level performance. In this paper, we present a novel web agent, Agent-E [†]. This agentic system introduces several architectural improvements over prior state-of-the-art web agents, such as hierarchical architecture, self-refinement, flexible DOM distillation, and *change observation* to guide the agent towards more accurate performance. Our Agent-E system without self-refinement achieves SOTA results on the WebVoyager benchmark, beating prior text-only benchmarks by over 20.5% and multimodal agents by over 16%. Our results indicate that adding a self-refinement mechanism can provide an additional 5.9% improvement on the Agent-E system without self-refinement. We then synthesize our learnings into general design principles for developing agentic systems. These include the use of domain-specific primitive skills, the importance of state-sensing and distillation of complex environmental observations, and the advantages of a hierarchical architecture.

## 1 INTRODUCTION

Recent studies indicate that generative AI and automation tools could handle 60-70% of an employee's tasks (Chui et al., 2023). By reducing cognitive load, saving time, and optimizing workflows, these tools can potentially contribute between $2.6 trillion and $4.4 trillion to global GDP (Chui et al., 2023). With the rise of digital jobs and advancements in the reasoning abilities of large language models (LLMs), these models are increasingly being integrated into autonomous systems for a variety of tasks. LLM-agents can be seen in applications like software engineering tasks (Jimenez et al., 2023; Huang et al., 2023a; Zhang et al., 2024b; Schick et al., 2023), personal device control (Yan et al., 2023; Wu et al., 2024; Zhang et al., 2024a), and web navigation (He et al., 2024; Zhou et al., 2023; Putta et al., 2024b; Lutz et al., 2024b). However, while these agents have demonstrated promising results in some areas, their performance in web automation remains limited.

Several unique challenges make planning difficult in a web navigation context. First, websites are represented in HyperText Markup Language (HTML) Document Object Models (DOMs), which organize elements in a nested format. These lengthy, dynamic text-based representations complicate the identification of key elements in the observation space (Lutz et al., 2024b). Furthermore, DOMs often exceed the context windows of current state-of-the-art LLMs. Second, while humans can naturally execute complex web tasks, agents require careful, multi-step planning. Even a simple task, like a *Google search (e.g. clicking the search bar, typing each key, and pressing enter)*, involves multiple fine-grained actions. Lastly, current state-of-the-art web agents remain error-prone and unreliable for deployment, underscoring the need for further advancements in this area to create a more reliable system (Wornow et al., 2024; He et al., 2024; Zhou et al., 2023).

In this paper, we introduce Agent-E, a state-of-the-art web agent capable of performing complex web-based tasks. Our system presents several design elements that elevate challenges faced by prior

---

[†]Implementation available at: `https://anonymous.4open.science/r/Agent-E-7E43`

web navigation systems. Central to Agent-E are three LLM-powered components: the planner agent browser navigation agent, and verification agent. The planner agent is responsible for high-level planning and task management. It breaks down the user task into a sequence of high-level tasks and delegates them to the browser navigation agent. The browser agent then plans and executes the lower-level steps necessary to complete the delegated task. This tiered system breaks down the planning into fine-grained actions that are more manageable tasks; this insulates the planner agent from the low-level details of the observation space. To further improve the interpretability of DOMs, our system utilizes different DOM distillation techniques. These techniques emphasize features in the DOM relevant to completing an action to prevent an LLM agent from becoming overwhelmed with the difficult observation space. In addition, our system employs a validation agent at the end of each task. This validation agent provides feedback on incomplete tasks, leading to a self-correcting system.

Using our proposed system, we demonstrate that web agents can achieve state-of-the-art performance on realistic web navigation tasks without additional supervision. By combining our hierarchical system with DOM distillation techniques, we attain a new state-of-the-art 73.1% result on the WebVoyager benchmark (He et al., 2024), which is 20.5% higher than previous text-only web agents (Lutz et al., 2024b) and 16% higher than previous multi-modal web agents (He et al., 2024). Additionally, we achieve a 5.9% boost in performance using a self-refinement mechanism.

## 1.1 CONTRIBUTIONS

- We introduce a novel hierarchical architecture for web agents that enables the execution of more complex tasks through a clear separation of roles and responsibilities between a planner agent and a browser navigation agent.
- We introduce two novel components in Agent-E, a flexible DOM distillation approach where the browser navigation agent selects the most suitable DOM representation given the task, and the concept of *change observation*, a Reflexion-like paradigm (Shinn et al., 2024), where the agent monitors state changes after each action and receives verbal feedback to enhance awareness and performance.
- We propose a self-refinement mechanism for web navigation that enables workflows to be verified and self-corrected during failures, leading to more reliable web navigation workflows where failures can be detected.
- We report detailed end-to-end evaluations of Agent-E on the WebVoyager benchmark and show that it achieves new state-of-the-art results with a 73.1% success rate without self-refinement. Our system shows consistent improvement over different modalities, showing over a 20.5% improvement for text-based agents and 16% improvement for multi-modal. And another 5.9% boost in performance on a subset of WebVoyager tasks when self-refinement is added.

In Section 2, we give a lower-level view of Agent-E and how each of the design choices is implemented. Then in Section 3, the web navigation evaluation procedure and results are presented. We synthesize our findings into a list of design principles in Section 4. We provide related work and summarize our findings in Section 5 and 6.

## 2 AGENT-E: SYSTEM DESCRIPTION

Agent-E is built using Autogen, the open-source programming framework for building multi-agent collaborative systems (Wu et al., 2023b) and Playwright* for browser control. Our system simplifies complex, long-horizon planning for web navigation workflows. Agent-E hierarchical system is composed of three LLM-powered agents: Planner, Browser Navigation Planner, Validation Agents, and one execution component. Each component plays an integral role in the system's successful and reliable workflow execution.

To manage the different granularity of sub-tasks necessary to complete a full workflow, our system is split into a hierarchy: 1) high-level planning, which is performed by the planner agents, and 2) low-level planning and execution, which is handled by the browser navigation planner and executor.

---

*https://playwright.dev/

Given a new user task, the planner agent decomposes the task into a sequence of high-level steps. Then throughout the workflow, the planner agent delegates the execution of each high-level step to the browser navigation subsystem and adapts to the plan based on the observation from the browser navigation subsystem. Finally, once the planner indicates the workflow is completed, the validation agent verifies the workflow. During workflow failures, the validation agent returns feedback to the planner agents and prompts them to correct its workflow. The self-refinement mechanism is further explained in Section 3.

To tackle the challenges of large observation spaces and fine-grained action space in browser interactions, we introduce the notion of skills, a set of predefined actions the agents can execute. These predefined can be associated with the execution of actions, or related to sensing the current observation space.

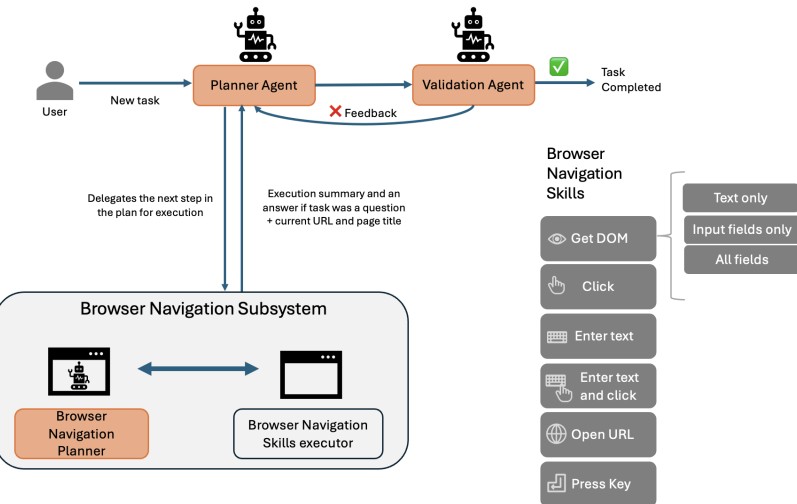

Figure 1: A high-level architecture of Agent-E

Our browser navigation agent has a set of foundational skills for observing a simplified observation space or controlling the browser. This agent uses the skills available to perform the sub-task and return a summary of actions it took to perform the task and/or answer the planner if the task was a question (See Table 2). Next in Section 2.1, we introduce our set of pragmatic predefined skills which can significantly simplify complex fine-grained web navigation tasks to an agent.

Lastly, our sensing skill relies on *change observations*, the ability to monitor element attributes (e.g., *aria-expanded*) or detect the addition of new elements (e.g., using the Mutation Observer Web API). This enables immediate detection of DOM updates following an action execution, which is particularly beneficial for highly dynamic pages, such as flight booking websites (e.g. Figure 4). A more detailed explanation of our implementation is provided in Appendix E.

## 2.1 SKILLS DESIGN & DOM DISTILLATION FOR BROWSER NAVIGATION AGENT

There are two key novel components in skills design used in Agent-E.

- Sensing Skills & DOM Distillation: Agent-E supports three different DOM distillation techniques (*text only, input fields, all fields*) that allow the browser navigation agent to choose the approach best suited for the task (see Figure 2). If the task is to summarize information on a page, it can simply use *Get DOM* with *text_only* content type. If the task is to identify and execute a search on a page, it can use the content type *input_fields*. If the task is to list all the interactive elements on a page, it can use *all_fields*. This optimizes the information available to the agent and prevents the problems associated with noisy DOM. Another key aspect is that our DOM distillation techniques for *all_fields* and *input_fields* attempt to preserve the parent-child relationship of elements wherever possible and relevant. This is unlike some previous implementations which use a flat DOM encoding (e.g. Lutz

et al. (2024b)). Further, to make identifying and interacting with HTML elements easier, Agent-E injects a custom identifier attribute (*mmid*) in each element as part of sensing, similar to Zhou et al. (2023) and He et al. (2024).

- Action Skills: All the action skills are designed to not only execute an action but also report on any change in state as an outcome of the action, a concept we call *change observation*. This is conceptually similar to the Reflexion paradigm (Shinn et al., 2024) which uses verbal reinforcement to help agents learn from prior failings. However, a key difference is that *change observation* is not directly associated with or limited to a prior failure. The observation returned can be any type of outcome of the action. For example, a *click* action may return a response *Clicked the element with mmid 25. As a consequence, a popup has appeared with the following elements*. Such detailed skill responses nudge the agent toward the correct next step.

| Skills | Input parameter | *Change Observation* during skill execution | Return |
|---|---|---|---|
| Get DOM | content type: text_only | None | Returns the innertext of body element of HTML DOM with some post processing. Ideal for text summarization and information extraction. |
| | Content type: input fields | None | Returns a json representation of specific HTML elements such as buttons, input fields and links in a page. Ideal for interacting with search fields or buttons. |
| | Content type: all fields | None | Returns a json representing the full page. Most complete representation of all elements, also most lengthy and noisy. |
| Click | Selector: identifier of the element to be clicked | Observe for DOM change events immediately following the click. | Returns a textual response indicating if click was performed and a summary of changes observed (if any). |
| Enter text | Selector: identifier of the element to enter text. | Observe for DOM change events during or following the text input. | Returns a textual response indicating if text input was performed and a summary of changes observed (if any). |
| Open URL | URL: The url to navigate | Web navigation | Page url and title of the new page |
| Press Key | Keys to press. (e.g. Submit, PageDown) | Observe for DOM change events immediately following the key press. | Returns a textual response indicating if keypress was performed and a summary of changes observed (if any). |

Figure 2: Skills registered to the Browser Navigation Agent for sensing and acting on the web page.

## 2.2 SELF-REFINEMENT

Our Agent-E system uses a self-refinement mechanism (Madaan et al., 2023) which allows the agent to self-correct incorrect workflows. We complement our planner and browser navigation agents with a validation agent that assesses the completion of the task. In cases where a task remains incomplete, the agent leverages the validator's feedback to revise its strategy and reattempt the task. The high-level mechanism illustrated in Figure 3, will allow the agent to self-correct in detected failures. Note the validation agent is only invoked once the planner agent finishes its workflow.

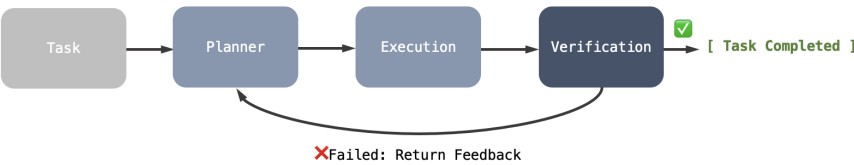

Figure 3: Self-refinement workflow.

Building on the concepts of LLM-as-a-judge (Zheng et al., 2024b) and self-critique mechanisms, we utilize LLMs to form validation agents. Prior work has suggested that providing multimodal observations leads to the best performance in LLM-based planners (Koh et al., 2024; He et al., 2024). Thus, we implement and test different modalities of validator agents: text and vision. The implementation details and investigation of our validator(s) can be found in Appendix B.

# 3 EVALUATION

In this section, we test and demonstrate that our Agent-E system outperforms other web agent systems with the use of no additional supervision. Our results indicate that our hierarchical architecture, sensing and action space, and use of self-refinement are able to make better use of LLM context windows for planning.

**WebVoyager Benchmark** WebVoyager (He et al., 2024) is a recent web agent benchmark that consists of web navigation tasks across 15 real websites (e.g. Amazon, Google Flights, Github, Booking.com). Each website has about 40-46 tasks resulting in a benchmark dataset of 643 tasks. We chose WebVoyager since it covers a diverse range of tasks across dynamic, live websites that are representative of real-life use cases for web navigation. In contrast, alternative benchmarks either focus narrowly on a single task domain (Yao et al., 2022), lack dynamic website observations (Deng et al., 2023), or rely on custom websites with significantly less complex DOM structures than those found in real-world environments (Liu et al., 2018; Zhou et al., 2023).

**Experimental Details** The entire benchmark was divided among 5 human annotators. For each task, an annotator was instructed to classify the task as *pass* or *fail* along with a textual reason in case of failures. A task is considered complete only if the agent successfully finishes all parts of the instructed task and remains on the designated website. *Overall accuracy* measures the percentage of times the validator's label and human annotator's labels match. To remain consistent with prior work benchmarks on WebVoyager (He et al., 2024), we utilize GPT-4-Turbo (gpt-4-turbo-preview) as a planner and browser navigator in our Agent-E implementation. And for the validation agent, we employ GPT4-Omni(gpt-4o).

## 3.1 AGENT-E SYSTEM RESULTS

In this section, we present quantitative results measuring Agent-E's performance on the WebVoyager benchmark. Table 1 shows the summary of the evaluation of Agent-E w/o Self-Refinement on WebVoyager. Due to limitations of annotator time, our results for Agent-E with self-refinement include 456 uniformly selected Web-Voyager tasks. Table 2 presents the evaluation of Agent-E on this subset of WebVoyager tasks.

| Publication | Task success rates on websites | | | | | | | |
|---|---|---|---|---|---|---|---|---|
| | Allrecipe | Amazon | Apple | Arxiv | Github | Booking | ESPN | Coursera |
| He et al. (2024) (text) | 57.8 | 43.1 | 36.4 | 50.4 | 63.4 | 2.3 | 28.6 | 24.6 |
| He et al. (2024) (multi) | 51.1 | 52.9 | 62.8 | 52.0 | 59.3 | 32.6 | 47.0 | 57.9 |
| Lutz et al. (2024b) (text) | 60 | 43.9 | 60.5 | 51.2 | 22.0 | **38.6** | 59.1 | 51.1 |
| Agent-E (text) | **71.1** | **70.7** | **74.4** | **62.8** | **82.9** | 27.3 | **77.3** | **85.7** |
| Publication | Task success rates on websites | | | | | | | |
| | Dictionary | BBC | Flights | Maps | Search | Hug.Face | Wolfram | Overall |
| He et al. (2024) (text) | 66.7 | 45.2 | 7.1 | 62.6 | 75.2 | 31.0 | 60.2 | 44.3 |
| He et al. (2024) (multi) | 71.3 | 60.3 | **51.6** | 64.3 | 77.5 | 55.8 | 60.9 | 57.1 |
| Lutz et al. (2024b) (text) | **86.0** | **81.0** | 0.0 | 39.0 | 67.4 | 53.5 | 65.2 | 52.6 |
| Agent-E (text) | 81.4 | 73.8 | 35.7 | **87.8** | **90.7** | **81.0** | **95.7** | **73.1** |

Table 1: Evaluation of Agent-E on 642 tasks WebVoyager across multiple websites..

Agent-E without self-refinement, completed 73.1% of the tasks, outperformed the text-only web agent WILBUR (Lutz et al., 2024b) by 20.5% and multi-modal web agent (He et al., 2024) by 16%, thus highlighting the importance of a system which 1) can break down tasks hierarchically and 2) utilizes DOM distillation for simplified sensing of a complex observation space. Additionally, we indicate the benefits of utilizing a self-refinement mechanism. We observe another 5.9% improvement, across the board for both modalities, when self-refinement is added to our Agent-E system – reaching a performance of 81.2% on the subset of WebVoyager tasks (for which Agent-E without self-refinement had a task success rate of 75.3%).

Although the overall performance of modality has little variance (i.e. 80.9%-81.2%), the task-specific performance is highly modality dependent. For example, for Google Flights, text validation

| System configuration | Task success rates on websites | | | | | | | |
|---|---|---|---|---|---|---|---|---|
| | Allrecipe | Amazon | Apple | Arxiv | Github | Booking | ESPN | Coursera |
| Agent-E w/o Self-Refinement (text) | 71.0 | 73.3 | 77.4 | 67.7 | 83.8 | 25.8 | 79.3 | 93.3 |
| Agent-E (text) | **77.3** | 86.4 | **95.5** | **90.9** | **100.0** | 27.3 | 72.3 | 86.4 |
| Agent-E (vision) | 73.7 | **91.3** | 86.4 | 66.7 | **100.0** | **41.2** | **100.0** | **95.2** |
| Publication | Task success rates on websites | | | | | | | |
| | Dictionary | BBC | Flights | Maps | Search | Hug.Face | Wolfram | Overall |
| Agent-E w/o Self-Refinement (text) | 80.7 | 74.2 | 37.9 | **83.3** | 93.6 | **87.1** | **100** | 75.3 |
| Agent-E (text) | **95.5** | **90.9** | 57.1 | 76.2 | **95.2** | 76.5 | 90.2 | 80.9 |
| Agent-E (vision) | **95.5** | 80.95 | **85.7** | 76.2 | 76.2 | 63.2 | 77.3 | **81.2** |

Table 2: Evaluation of Agent-E on a subset of 458 WebVoyager tasks across multiple websites.

achieves 57.1% while vision achieves 85.7%. Tasks that are primarily text-based and performed on simple websites tend to perform best with the text validator (e.g., Google Search, Arxiv, Hugging Face, WolframAlpha). In contrast, websites that are highly dynamic with complex DOMs perform better with the vision validator (e.g., Google Flights and Booking.com). Notably, Booking.com shows performance gains of over 13% using vision over text.

Moreover, it is important to note that WILBUR (Lutz et al., 2024b) uses task and website-specific prompting, while He at al. (He et al., 2024) uses vision for observing the page. In contrast, Agent-E is a planner agent and browser navigation is, a text-only web agent that does not employ any task or website-specific instructions. The vision version of Agent-E is only reflected in the choice of the validation agent. This suggests that there is likely room for further improvement in Agent-E using website/task-specific strategies and vision.

### 3.1.1 TASK COMPLETION TIME

In Table 3, we can see the amount of time taken to complete each workflow with and without-refinement. The average run time of Agent-E w/o refinement is $\sim$ 3 minutes while refinement is $\sim$ 6 minutes. Although the self-refinement mechanism was able to show improvement in overall performance, this process is time-consuming, only allowing the agent to correct its workflow at the end of each run. This indicates the cost associated with the outcome-based self-refinement process.

| | Allrecipe | Amazon | Apple | Arxiv | Github | Booking | ESPN | Coursera |
|---|---|---|---|---|---|---|---|---|
| W/o Self-Refinement | 140 | 282 | 132 | 156 | 161 | 299 | 450 | 115 |
| Agent-E (text) | 124 | 659 | 272 | 441 | 157 | 838 | 269 | 297 |
| Agent-E (vision) | 322 | 435 | 307 | 307 | 399 | 743 | 569 | 1266 |
| | Dictionary | BBC | Flights | Maps | Search | Hug.Face | Wolfram | Overall |
| W/o Self-Refinement | 106 | 108 | 248 | 120 | 90 | 147 | 69 | 173 |
| Agent-E (text) | 75 | 166 | 288 | 398 | 373 | 159 | 119 | 319 |
| Agent-E (vision) | 118 | 161 | 452 | 236 | 213 | 150 | 165 | 376 |

Table 3: Average Time (Seconds) Per Task Execution on 458 WebVoyager tasks.

For the case of Agent-E without self-refinement, we can see that easier tasks take less time to complete. For example, tasks like Dictionary, Maps, and Search, which all have high success rates, also have some of the lowest run times. Additionally, results on the task completion times of Agent-E without refinement are provided in the Appendix A.

### 3.2 QUALITATIVE ANALYSIS

In this section, we present qualitative results with concrete examples showing how different design choices made in Agent-E help perform complex web tasks.

### HIERARCHICAL PLANNING HELPS ERROR DETECTION AND RECOVERY

The hierarchical architecture allows easy detection and recovery from errors. The planner agent is prompted to perform verification (by asking questions or asking for confirmation) as part of the plan whenever necessary. Shown in Appendix F, Figure 7 shows an example instance where the planner agent asks the browser navigation agent for more information (i.e., *list the search results*), and from

the response (i.e., *there are no specific search results*) identifies that it may have made an error by making the search query too focused. In the example, the planner creates a new plan of action for performing the task. Another common pattern in the evaluation was the planner's ability to detect errors and easily backtrack to a previous page to continue execution. Given that the planner has the URL of the page at each step available to it, it allows the planner to effortlessly backtrack to a previous page by adding it as a step in the plan (e.g., *navigate to the search result page using the <url>*). Refer to Appendix C for an ablation comparing the hierarchical system with a simpler single-agent system.

## THE NEED FOR MULTIPLE DOM OBSERVATION METHODS

Typical HTML DOMs can be extremely large (e.g., the YouTube homepage with all DOM elements and attributes is about 800,000 tokens). Thus, it is important to denoise and encode the DOM such that only task-relevant information is presented to the LLM. However, information relevant to a given task is very dependent on the task at hand. Some tasks may only need a complete textual representation (e.g., *summarise the current page*), and some tasks may only need input fields and buttons (e.g., *search on google*). On the other hand, more exploratory tasks may need a complete representation of the page (e.g., *what elements are on this page*).

Most previous web agents have used a single DOM representation, e.g. (Zhou et al., 2023) used an accessibility tree,(He et al., 2024) used screenshots and (Lutz et al., 2024b) used direct encoding and denoising of the HTML DOM. However, in our view, there is no single DOM observation method that suits all the tasks. Thus, Agent-E supports three different DOM representation methods *text_only, input_fields, all_fields*. This allows Agent-E to flexibly select the DOM representation that it feels is best suited for the task. Also, this allows Agent-E to fall back to different representations, when one representation unexpectedly does not work well. There were numerous examples in our benchmark where these multiple DOM representations were useful. Appendix A: Figure 6 illustrates an example where Agent-E adaptively uses *all_fields* DOM representation for interaction and *text_only* for summarization. Refer to Appendix D for quantitative evaluation comparing the flexible DOM distillation and directly using the accessibility tree.

## CHANGE OBSERVATION HELPS GROUNDING

Change observation is a technique where each action execution is accompanied by observation of state changes, and this is returned via linguistic feedback to the LLM. A typical scenario where this is useful is when the browser navigation agent tries to click on a navigation item (e.g., *click on the soccer link on ESPN.com*), and instead of navigating to the relevant section, the page instead opens a popup menu that requires further selection. In this example, the interaction is not yet complete (since completion requires clicking a popup link or selecting a drop-down entry, respectively), but the browser navigation agent may assume it is complete. With change observation, the *click* skill will return feedback to the LLM that *as a consequence of the click, a menu has appeared where you may need to make further selection*. See Figure 4 in Appendix F for an example.

The purpose of change observation is to provide linguistic feedback to the LLM on whether the action led to any tangible change in the environment, to inform subsequent actions. We also envision efficiency improvements if the change observation can return a list of elements so that LLM can make subsequent selections without again using the *Get DOM* skill to observe the state of the DOM.

Change observation is adjacent to the concept of Reflexion (Shinn et al., 2024). However, there are nuanced differences between the two. The Reflexion technique provides feedback on a prior failure, by using an LLM to analyze the scalar 'success/failure' signal based on an action and current trajectory. In contrast, change observation is not a binary signal and instead observes the change in the environment as a consequence of an action (e.g. new elements added to DOM, pop-up expanded, etc). Change observation is implemented using mutation-observer API to observe the consequence of an action and provide linguistic feedback of actions to help the system be better aware of the new state of the environment, and nudge the system towards the correct next action.

# 4 AGENT DESIGN PRINCIPLES

In this section, we synthesize our learnings from the development and evaluation of Agent-E into a series of agent design principles. We believe these principles can be generalized beyond the domain of web automation.

1. **Design with a Core Set of Primitive Skills to Enable Versatile Use Cases**: An ensemble of well-crafted foundational skills can serve as a building block to support more complex functionalities. LLMs can effectively combine these skills to unlock a broad range of use cases. These skills should be domain-dependent; in the case of Agent-E, these primitive skills were *click, enter text, get DOM, Open URL* and *Press Keys*. These are only a subset of potential user actions on a page (e.g. we do not support *drag, double click, right-click, tab management*, etc). We consider the skills enabled in Agent-E enough for the vast majority of general web automation tasks. However, websites with specialized interaction patterns (e.g. right-click to select functionality) may benefit from additional skills. Examples of prior related work leveraging domain-specific primitive skills include (Irpan et al., 2022; Nakano et al., 2022; Lutz et al., 2024b) among several others, highlighting the generality of the design principle.

2. **Adopt Hierarchical Architectures for Managing Complex Task Execution**: The idea of using hierarchical AI planning for complex tasks has existed for decades (Tate, 1977; Nau et al., 1991; Marthi et al., 2007); see Russell & Norvig (2009) for details. In agents with multiple LLM-based components, a hierarchical architecture excels in scenarios where tasks can be decomposed into sub-tasks that need to be handled at different levels of granularity. In the case of Agent-E, this allows the high-level planner to be agnostic of browser-level details. Additionally, it aids in the identification of tasks that can be executed in parallel, leading to performance enhancements. It also supports the development and improvement of various components in isolation. Note that hierarchical architectures may not always be necessary. In the case of Agent-E, if all we had to support were simple tasks like navigating to specific URLs or performing a web search, a hierarchical architecture might be over-complicated, and a simpler architecture may have sufficed.

3. **Domain-Specific State Processing Improves Efficiency and Accuracy**: Depending on the domain, there may exist a large amount of environmental information, much of which is irrelevant. An example is HTML DOMs for websites which may have hundreds of thousands of tokens. This may lead to suboptimal LLM performance, especially for sequential decision-making tasks. Agent-E employs a variety of domain-specific processing and sensing techniques to distill only task-relevant data. These include multiple DOM filtering approaches that the agent adaptively uses given the task requirements. Removing as much noise as possible from the environment before the system begins processing, is a crucial requirement while building such agentic systems.

4. **Integrate Linguistic Feedback to Summarize State Changes**: Agent-E's actions change the state of the page, often in complex ways. We have found that, rather than relying on the filtered DOM alone, explicitly detecting and summarizing state changes through linguistic feedback enables the agent to more effectively understand the consequences of an action (e.g., *a dialog box appeared as a consequence of the click action*). *Change observation* helps refine the agent's subsequent actions by providing a clear narrative of cause and effect, and also improved awareness of the environment. This idea is also applicable in other contexts, for example, in use cases such as desktop automation or automation in the physical space (e.g robot control). Examples of systems that use related ideas in other domains include (Wang et al., 2023c; Song et al., 2023; Wang et al., 2023a) among others. Descriptive logging and tracking are highly beneficial in agentic systems.

5. **Leverage Past Experience**: For agentic systems to be adopted widely, they need to achieve close to human-level performance. One approach is for agents to routinely reflect and learn from their past experiences. Our Agent-E system introduces the use of self-refinement for web automation. The 5.9% boost in performance achieved by this mechanism shows that agents are capable of identifying and self-correcting their mistakes throughout a single task execution. A more efficient approach to leveraging past workflows is to establish offline workflows that routinely analyze, reflect on, and aggregate past tasks and human demonstrations to convert them to more classical automation workflows. These automated

workflows could then be re-triggered upon a new task if it matches a workflow that has been encountered in the past, with the exploratory agentic approach used only as a fallback. This would enable faster and cheaper task completion, which should be a primary requirement of agentic systems. Other examples of leveraging past experience can be found in prior work on self-improving systems e.g. (Zelikman et al., 2022; Hosseini et al., 2024; Wang et al., 2023c) and others.

6. **Balance Between Generic and Task-Specific Agent Design**: Generic agentic systems by definition can perform a wide range of tasks. However, in many practical implementations, a more focused set of capabilities may be desirable. For example, Agent-E is a generic web agent that can perform a wide range of tasks on the internet but is not necessarily optimized for any specific task. It would be possible to optimize Agent-E for specific types of tasks (e.g., form filling) or specific websites (e.g., Atlassian Confluence pages) to achieve significantly higher performance. Depending on the use case, an optimized agent may suit better for certain workflows than a generic version.

## 5 RELATED WORK

**LLM-based Planning and Reasoning**   Over the last few years, Large language models (LLMs) have excelled in text generation, code generation, and the generation of multistep reasoning. This has spurred the use of LLMs to solve multi-step reasoning and planning problems. The many variants of 'chain-of-thought' techniques (Wei et al., 2022; Chu et al., 2023) encourage the LLM to produce a series of tokens with causal decoding that drive toward the solution of problems in math, common-sense reasoning and other similar tasks (Chowdhery et al., 2022; Fu et al., 2023; Li et al., 2023; Mitra et al., 2024). With tool-usage for sensing and acting, LLMs have also been used to drive planning in software environments and embodied agents e.g., (Baker et al., 2022; Wang et al., 2023a;c; Irpan et al., 2022; Bousmalis et al., 2023; Wu et al., 2023a; Bhateja et al., 2023). Finally, there has been related work investigating the limits of LLMs when it comes to planning and validation. For examples of negative results, see (Valmeekam et al., 2023b; Momennejad et al., 2023; Valmeekam et al., 2023a; Huang et al., 2023b; Kambhampati et al., 2024) among others. In this paper, we investigate multi-step planning for specialized web agents. We find that domain-specific techniques including sensing (through DOM distillation and change-observation), hierarchical planning (with a low-level browser agent), and multimodal self-refinement, are crucial for state-of-art performance.

**Specialized Agents for Repetitive Tasks**   Beyond the examples above, and as described in Section 1, there has been much recent interest in building specialized agents for the web (Zheng et al., 2024a; He et al., 2024; Lutz et al., 2024b) and on device (Bai et al., 2024; Wen et al., 2024). Also related is recent work on building agentic workflows to replace robotic process automation (Wornow et al., 2024). Further, the work on building agents and training language models for API usage is also related, given that many software tasks and workflows involve the use of APIs; examples include (Hosseini et al., 2021; Patil et al., 2023; Qin et al., 2024) and many more. As described in Section 1, our proposed web agent employs multiple novel ideas that enable it to achieve state-of-art performance on realistic web navigation tasks, significantly outperforming previous specialized web agents.

**Hierarchical Planning**   The notion of hierarchical AI planning has been around for five decades or more. Instead of planning directly in the space of low-level primitive actions, planning in a space of 'high-level actions' constrains the size of the plan length (and hence the size of the planning space), which can result in a more effective and efficient search. Examples from prior work include (Tate, 1977; Nau et al., 1991; Marthi et al., 2007) and many more; see Russell & Norvig (2009) for more details. Also related is the use of temporal abstractions in planning and reinforcement learning, for example, the use of options in (Sutton et al., 1999; Bacon et al., 2017). In recent years, multiple papers have proposed the use of hierarchical planning for solving tasks in complex environments or with embodied agents; examples include (Wang et al., 2022; Irpan et al., 2022) and others. In this paper, we introduce a hierarchical architecture for web agents where responsibility for planning and execution of complex web tasks is separated between a planner agent and a browser navigation agent. We show hierarchical planning is a promising solution for long-horizon planning in web navigation.

**Self-Improving Agents** Recent research has focused on enhancing the capabilities of LLMs during training or inference without additional human supervision (Wei et al., 2022; Chen et al., 2023; Wang et al., 2023b; Kojima et al., 2023). Techniques like chain-of-thought prompting and self-consistency, as used in Huang et al. (2023b), aim to generate higher-quality outputs. Other methods, such as Self-refine (Madaan et al., 2023), Reflexion (Shinn et al., 2023), and REFINER (Paul et al., 2024), focus on iterative refinement of outputs using actor-critic pipelines or task decomposition. These approaches have been successfully applied to web agents, improving the performance of LLMs in web automation tasks (Putta et al., 2024a; Pan et al., 2024; Lutz et al., 2024a). In this paper, we design and evaluate three different auto-validators, and use these to create self-refinement mechanisms for our web agent. Our results indicate that self-refinement, using our text- and vision-based auto-validators, shows notable additional gains in web navigation tasks.

## 6 CONCLUSION

This paper introduced Agent-E, a web agent that significantly advances the ability to handle complex web navigation tasks. Web-based automation faces key challenges such as the complexity of DOM interpretation and long-horizon task planning. Agent-E addresses these with flexible DOM distillation techniques to focus on relevant content, hierarchical task management to reduce error-prone low-level decisions and a self-refinement mechanism that allows the agent to correct its workflow without human intervention. Our evaluation of the WebVoyager benchmark demonstrates Agent-E's ability to overcome these web navigation challenges. With a 73.1% success rate, Agent-E without self-refinement sets a new state-of-the-art for web agents, surpassing prior text-based and multi-modal systems by 20.5% and 16%, respectively. We observe another 5.9% improvement when self-refinement is added to this system. We presented our learnings in the form of eight general design principles for developing agentic systems that can be applied beyond the realm of web automation.

Although Agent-E presents state-of-the-art results on web navigation tasks, there are several key observations and space for improvement. First, unlike prior state-of-the-art agents from He et al. (2024), our planning and browser navigation agent is not multimodal. Transitioning the browser navigation agent to handle multi-modal observations may improve its sensing capabilities. Second, although self-refinement shows the best performance, this outcome-based refinement system comes at a cost (i.e. requiring tasks to take 1.2-2x longer to complete). Lastly, we observed that different modalities of validation agents perform best for different tasks. This highlights the need for task-specific validation systems. In conclusion, Agent-E's novel approach effectively tackles key challenges in web navigation, offering a robust, adaptable framework that advances agentic systems in web automation and beyond. While task completion times can still be optimized, Agent-E provides a significant leap forward in agent performance and reliability.

### ETHICS STATEMENT

As web agents like Agent-E move beyond research prototypes, they can raise important ethical concerns. First, web agents that operate on a personal device may introduce privacy issues for the user. These agents may have access to user sensitive information including passwords and financial data. Second, such agents, if used by a malicious user, could potentially be used for harmful purposes like sending spam and unauthorized web scraping. Thirdly, the widespread deployment of web agents could violate websites' terms of service. While our research advances the technical capabilities of web agents, we recognize the critical importance of understanding failure modes and potential risks before real-world deployment. We acknowledge that benchmark performance alone is insufficient for ensuring safe deployment. Future work must establish robust security frameworks, access controls, and oversight mechanisms before web agents can be safely entrusted with user data and credentials. We emphasize that human oversight remains essential for deploying these systems responsibly.

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

# A  ADDITIONAL RESULTS: AGENT-E WITHOUT SELF-REFINEMENT

This section presents an additional quantitative evaluation of Agent-E without Self-Refinement, which was tested on WebVoyager. We report three additional measures relevant to the comprehensive evaluation of web agent and understanding of their practical implementation readiness.

- Self-aware vs. Oblivious failure rates: Detecting when the task was not completed successfully is of utmost importance since it can be used for enabling a fallback workflow, to notify the user of failure, or used as an avenue to gather human demonstration for the same task. Self-aware failures are failures where the agent is aware of their own failure in completing the task and responds with a final message explicitly stating so, e.g. *I'm unable to provide a description of the first picture due to limitations in accessing or analyzing visual content.* or *'Due to repeated rate limit errors on GitHub while attempting to refine the search...'.* The failures could be due to technical reasons or an agent deeming the task unachievable since it could not complete the task after repeated attempts. On the other hand, oblivious failures are cases where the agent wrongly answers the question or performs the wrong action (e.g. adds the wrong product to the cart or provides the wrong information). For mainstream utility, oblivious failures should be as minimal as possible. For the current evaluation, failures were categorized as self-aware and oblivious failures by manual annotation. However, it would be trivial to employ an LLM critique to automatically do the same task, similar to Wornow et al. (2024).

- Task completion times: The average time required to complete the task, across websites for failed and successful tasks.

- Total number of LLM calls: The average number of LLM calls (both planner and browser navigation agent) that were required to perform the task. This includes both successful and failed cases.

| | Allrecipe | Amazon | Apple | Arxiv | Github | Booking | ESPN | Coursera |
|---|---|---|---|---|---|---|---|---|
| Failure modes | Agent-E Error Analysis on Websites | | | | | | | |
| Overall failures % | 28.9 | 29.3 | 25.6 | 37.2 | 17.1 | 72.7 | 22.7 | 14.3 |
| Self-aware failures % | 17.8 | 14.6 | 9.3 | 18.6 | 12.2 | 4.5 | 13.6 | 4.8 |
| Oblivious failures % | 11.1 | 14.6 | 16.3 | 18.6 | 4.9 | 68.2 | 9.1 | 9.5 |
| TCT | Agent-E Avg. Task Completion Times (seconds) | | | | | | | |
| TCT (Success) | 116 | 286 | 122 | 137 | 104 | 183 | 187 | 119 |
| TCT (Failed) | 196 | 246 | 200 | 176 | 384 | 317 | 387 | 177 |
| LLM Calls | Agent-E Avg. Number of LLM calls | | | | | | | |
| Total | 22 | 23.1 | 21.5 | 25.5 | 21.5 | 36.4 | 24.0 | 25.5 |
| Planner | 6.5 | 6.4 | 5.9 | 6.9 | 5.4 | 6.6 | 6.3 | 6.3 |
| Browser Nav Agent | 15.5 | 16.7 | 15.6 | 18.6 | 16.1 | 29.8 | 17.7 | 19.2 |

Table 4: Evaluation of Agent-E without Self-Refinement on WebVoyager.

| | Dictionary | BBC | Flights | Maps | Search | Hug.Face | Wolfram | Overall |
|---|---|---|---|---|---|---|---|---|
| Failure modes | Agent-E Error Analysis on Websites | | | | | | | |
| Overall failures % | 18.6 | 26.2 | 64.3 | 12.2 | 9.3 | 19.0 | 4.3 | 26.9 |
| Self-aware failures % | 16.2 | 9.6 | 57.1 | 12.0 | 4.6 | 14.3 | 2.1 | 14.1 |
| Oblivious failures % | 2.4 | 16.6 | 7.1 | 0 | 4.6 | 4.7 | 2.1 | 12.8 |
| TCT | Agent-E Avg. Task Completion Times (seconds) | | | | | | | |
| TCT (Success) | 98 | 105 | 244 | 127 | 106 | 140 | 68 | 150 |
| TCT (Failed) | 136 | 110 | 234 | 177 | 135 | 167 | 94 | 220 |
| LLM Calls | Agent-E Avg. Number of LLM calls per Task | | | | | | | |
| Total | 22.0 | 21.3 | 53.8 | 22.9 | 19.4 | 22.8 | 14.5 | 25.0 |
| Planner | 6.6 | 6.0 | 11.4 | 5.8 | 5.6 | 6.2 | 4.4 | 6.4 |
| Browser Nav Agent | 15.4 | 15.3 | 42.2 | 17.0 | 13.7 | 16.6 | 10.15 | 18.6 |

Table 5: Evaluation of Agent-E on WebVoyager without Self-Refinement (Contd.)

**LLM Calls**   On an average it took 25 LLM calls to execute a task (6.4 calls by the planner and almost 3 times as much by the browser navigation agent). The average number of LLM calls per website, as expected, is consistent with task completion times. See Tables 4 and 5 in Appendix B for detailed analysis on LLM calls.

**Task Completion Times**   On average, it took significantly longer for completion when the task was a failure, versus on successful tasks (an average of 220 seconds on failed tasks vs 150 seconds on successful tasks, in our experiments). The longer duration for failed tasks is expected, since given a difficult task, Agent-E may try multiple approaches to complete the task before giving up on it. There were also significant differences in task completion times across websites (e.g., 68 seconds to successfully complete a task in WolframAlpha vs. 286 seconds in Amazon), reflecting the differences in task and website complexity. See Tables 4 and 5 in Appendix B for detailed analysis.

**Self-aware vs Oblivious failure rates**   We found that Agent-E was self-aware of failures, even without the self-validation process, for more than 52% of the failed tasks, i.e. it was obvious from Agent-E response that it could not complete the task (e.g. *I'm unable to provide a description of the first picture due to limitations in accessing or analyzing visual content.*). Typically, self-aware failures occur when the reason for failure are technical in nature (e.g., navigation issues, inability to extract certain information from DOM elements such as Iframes, canvas or images, inability to operate a button, anti-scraping policies employed by websites, inability to find the answer despite multiple attempts etc.).

On the other hand, oblivious failures are scenarios where Agent-E gave a response that was wrong. These are typically scenarios where the agent overlooks certain task requirements and provides an answer that only partially meets the requirements. These also stem from DOM observation issues (e.g., not being aware that the date got reset due to incorrect format in Google Flights) or issues in understanding website capabilities (e.g., not using advanced search capability when needed, or assuming search functionalities are perfect and every search result will completely satisfy the search requirements). Similar error modes were also observed by He et al. (2024) who classify them as agent *hallucinations*.

## B  AUTO-VALIDATOR RESULTS

This section demonstrates the effectiveness of three different LLM/VLM-based validation models. We implement our validator model using three different workflow representations:

1. **Task Log (Text)**: This method utilizes only the chat log between the planner agent and user proxy, containing the high-level actions and observations.

2. **Screenshots (Vision)**: This method employs a sequence of screenshots taken throughout the workflow execution.

3. **Screenshots & Final Response (Hybrid)**: This method combines a sequence of screenshots with the final text response provided by the planner agent.

Our validator is tested on workflows produced by Agent-E on the WebVoyager benchmark. Each workflow is labeled by human annotators to asses the accuracy. Our auto-validators are implemented with GPT4-Turbo for modalities with text only and GPT4-V for modes with vision.

Table 6 summarizes the performance of each modality of the validator. The Task Log (text) validator demonstrated the best performance, of $84.24\%$, with the hybrid validator performing similarly at $83\%$. The vision validator performed notably worse than the hybrid validator, indicating the importance of the agent's final answer in some tasks. However, between $17.68 - 19.67\%$ of tasks were labeled True Negatives.

|  | True Positive | True Negative | False Positive | False Negative | Validator Accuracy |
|---|---|---|---|---|---|
| Task Log (text) | 66.56 | 17.68 | 7.40 | 8.36 | **84.24** |
| Screenshots (vision) | 52.03 | 18.02 | 3.60 | 26.13 | 70.04 |
| Screenshot & Final Resp. (hybrid) | 63.33 | 19.67 | 5.00 | 12.00 | 83.00 |

Table 6: Confusion matrix and accuracies of validators.

The task-specific performance of the validators can be seen in Table 7. Although overall the text validator outperforms the vision validator, this section indicates there are tasks where the vision validator performs better. The Booking.com site has a notably difficult DOM, making it consistently one of the most challenging tasks for web navigation. For these tasks, the vision validator significantly outperforms the Task Log (text) validator. Additionally, the vision validator also performed notably better for Google Flight tasks. This website is highly dynamic and requires navigating widgets which are difficult to represent in the DOM. On the other hand, highly text-based tasks perform significantly better with some text modality (e.g., Google, Huggingface, Wolfram Alpha). This difference in task-specific performance highlights the benefit of having task-specific validators.

|  | Allrecipes | Amazon | Apple | Arxiv | BBC | Booking | Coursera | Dictionary |
|---|---|---|---|---|---|---|---|---|
| Task Log (text) | 82.22 | 75.61 | 79.07 | 88.37 | **92.86** | 69.77 | **95.24** | 93.02 |
| Screenshots (vision) | 80.00 | **77.50** | 80.49 | 88.37 | 90.48 | **85.00** | 92.86 | 95.24 |
| Screenshot & Final Resp. (hybrid) | **84.44** | 65.85 | **81.40** | **90.70** | 88.10 | 83.72 | 92.86 | **95.35** |
|  | ESPN | Github | Google | Maps | Flights | Hug.Face | Wolfram | Overall |
| Task Log (text) | 93.18 | 90.24 | **93.02** | 90.24 | 90.48 | 70.00 | **91.30** | **84.24** |
| Screenshots (vision) | **95.35** | **94.74** | 65.38 | 66.67 | 92.11 | 56.25 | 33.33 | 70.04 |
| Screenshot + Final Resp. (hybrid) | **95.35** | 87.80 | 92.31 | **90.48** | **95.12** | **75.00** | 90.48 | 83.00 |

Table 7: Validation agent accuracy by website.

## B.1 Validation Versions

In practice, our self-refinement mechanisms are implemented using GPT-4o, which serves as the backbone for generating reliable validation outputs. To evaluate the comparative performance of GPT-4o and GPT-4-Turbo-Preview as validation agents, we conducted a detailed analysis focused on accuracy and error rates across different metrics. The comparison aims to determine not only which model achieves higher accuracy but also which one offers a better balance of efficiency and reliability for deployment in real-world scenarios. The results of this evaluation are summarized in Table B.1.

|  | True Positive (%) | True Negative (%) | False Positive (%) | False Negative (%) | Validator Accuracy (%) |
|---|---|---|---|---|---|
| GPT-4-Turbo | 66.56 | 17.68 | 7.40 | 8.36 | 84.24 |
| GPT-4o | 70.51 | 14.51 | 9.20 | 5.77 | 85.02 |

Table 8: Confusion matrix and accuracies of validators.

As shown in Table B.1, GPT-4o achieves slightly higher accuracy compared to GPT-4-Turbo-Preview. Beyond accuracy, GPT-4o offers practical advantages such as faster execution times and greater cost efficiency, making it a more suitable option for large-scale deployment in computationally intensive pipelines. These benefits are particularly critical in scenarios where validation is performed repeatedly, as they contribute to reduced latency and operational expenses while maintaining high performance.

## C   Single Agent vs Hierarchical System

We conducted an evaluation of the single agent system and the hierarchical system (comprising of browser navigation agent and planner agent), using GPT-4-Turbo as the LLM for all agents in both configurations. The purpose of the evaluation was to better understand the trade-offs introduced by the hierarchical planner in terms of task success rates, task completion time, and number of LLM calls. We performed the evaluation using a subset of WebVoyager (75 tasks = 5 tasks randomly sampled tasks from each website * 15 websites). The results are presented below in Table 9.

The hierarchical system achieves higher task success rates. However, introduces increased computational overhead which is evident from longer task completion times and the number of LLM calls. The single-agent system, despite its lower computational cost, often struggles with tasks requiring multiple steps, exploration, or backtracking. Common failure modes included giving up prematurely if early attempts fail and providing incomplete answers without finishing the task in full. In contrast, the hierarchical system leverages its structured architecture to break down complex tasks into manageable sub-tasks, allowing the agents to handle long-horizon workflows more effectively, and allowing backtracking when a sub-task fails. Although this results in higher computational costs due to the additional steps required, it enables the system to complete these workflows successfully.

|  | Success Rate | TCT (seconds) | Avg. LLM Calls |
|---|---|---|---|
| Single Agent System (GPT-4-Turbo) | 48% | 68.2 | 9.2 |
| Hierarchical System (GPT-4-Turbo) | 70.6% | 170 | 29 |

Table 9: Performance Comparison of Agent-E Configurations

## D   FLEXIBLE DOM DISTILLATION ABLATION

A key distinction between Agent-E and other web agents (e.g. He et al. (2024) and Lutz et al. (2024b)) is that Agent-E supports multiple DOM observation techniques that the LLM can choose from, given the task at hand. Our DOM distillation method consists of three DOM observation techniques which can be selected by the Browser Navigation Agent depending on the task:

1. **all_fields:** This is the most comprehensive DOM representation, provided in JSON format. It starts with the Accessibility Tree (AXTree) of the webpage—a simplified version of the DOM that omits non-semantic elements like `<div>` tags used purely for styling. We the enrich this view with additional details, such as the names of HTML tags and inner text content where necessary. This representation is useful for tasks requiring detailed interaction with page elements.

2. **input_fields:** This is a subset of `all_fields` where only input fields and interactive elements from the DOM are returned. This strips away all the non-interactive text elements and allows the agents to use a much more succinct version of the DOM for purely interaction purposes.

3. **text_only:** This is a plain text view of the current page (gathered by using `body.innerText` in JavaScript of the current page). This will not have DOM identifiers to interact with screen elements but will have full text visible on the page. This is best suited for summarizing page content or answering specific questions from the page (e.g., *What is the price of iPhone 16?* or *Is this product waterproof?*). Answering such questions with `all_fields` is a lot more challenging since the information can be fragmented across multiple DOM fields and thereby multiple JSON nodes.

Prior work typically uses a single DOM observation method such as a simplified version of the HTML DOM based on heuristics (Lutz et al. (2024b)) or directly uses the accessibility tree (AxTree) of the current page (e.g. Zhou et al. (2023), He et al. (2024)).

To better understand the value provided by the flexible DOM distillation, we conducted an evaluation comparing Agent-E with flexible DOM distillation with a variant that directly uses AxTree. We performed the evaluation using a subset of WebVoyager (75 tasks = 5 tasks randomly sampled tasks from each website * 15 websites), the same as described in Appendix C.2. The results are presented below in Table 10. Note the experiment below is Agent-E without self-refinement.

Flexible DOM distillation significantly improves success rates (+16%) by tailoring observations to task-specific needs. Using AXTree directly is marginally faster since the AXTree enrichment steps we perform for 'all fields' and 'input fields' take some processing time (typically an additional 1-2 seconds per call depending on the complexity of the webpage). These findings emphasize the importance of adaptive DOM distillation in enhancing Agent-E's effectiveness across diverse web navigation tasks.

|  | Success Rate | TCT (seconds) | Avg. LLM Calls |
|---|---|---|---|
| Flexible DOM distillation | 70.6% | 170 | 29 |
| AXTree only | 54.6% | 161 | 37 |

Table 10: Performance Comparison of Agent-E Configurations

# E CHANGE OBSERVATION IMPLEMENTATION DETAILS

Identifying what has changed on a website as a consequence of an action is a non-trivial problem because websites are implemented using diverse approaches. For example, some websites dynamically add new elements to the Document Object Model (DOM) after an action. Other websites achieve similar effects by modifying properties like visibility, opacity, position or display styles of existing elements, without adding new ones. In Agent-E, we implement Change Observation using two complementary approaches: tracking changes in aria-expanded attribute and tracking new elements added using Mutation Observer.

**Tracking changes in aria-expanded attribute:** The ariaexpanded attribute is a standard accessibility feature that indicates whether a particular element (e.g., a menu or dropdown) is expanded or collapsed. By observing if aria-expanded changes from False to True, we can infer if the element has changed state (e.g. "Click action on the element [mmid=25] was performed successfully). As a consequence a menu has appeared where you may need to make further selections. Get all_fields DOM to complete the action." a relatively straightforward approach that tells the LLM that a menu is now open and likely further actions are needed. This method works effectively on websites that adhere to accessibility standards, regardless of how the underlying site is implemented. Figure 4 shows an example and the following steps describe how change observations for aria-expanded attribute is implemented in Agent-E:

1. LLM invokes an action skill (e.g. click on element with mmid 823)
2. Check if the element has an aria-expanded property and its value
3. Perform the click operation
4. Wait 100ms.
5. Check the new aria-expanded property and if it toggled from False to True.
6. If no, return a standard response: *Success. Executed JavaScript Click on element with selector: [mmid='823']*
7. If yes, return an additional message *Success. Executed JavaScript Click on element with selector: [mmid='823']. As a consequence a menu has appeared where you may need to make further selection. Get all_fields DOM to complete the action.*

**Using a DOM Mutation Observer:** Mutation observers are tools that monitor changes in the DOM, such as the addition or modification of elements. We use this mechanism to detect if new elements are added after an action. In our case, we listen to changes that relate to the addition of new elements (if developers of the website are using a different approach, e.g. toggling the visibility of existing elements, this will not return any changes). Before any action is invoked, we subscribe to a mutation observer on that page and listens to any changes during the skill execution and an additional 100ms. The mutation observer returns a list of new elements that were added and we return that list to the LLM with an additional message. The following steps describe how change observations for newly added elements is implemented in Agent-E.

1. LLM invokes an action skill (e.g. enter text "fake news detection model" on element with mmid 122)
2. Subscribe to DOM mutation observer for the full page
3. Perform the enter text operation
4. Wait 100ms
5. Unsubscribe the DOM mutation observer
6. Analyse if any new elements were added during this window.
7. If No, simply return a success message: *Success. Text "fake news detection model" set successfully in the element with selector [mmid='122']*
8. If new elements were added, return a short list of elements with the return message. In the above example, it would return: *Success. Text "fake news detection model" set successfully in the element with selector [mmid='122']. As a consequence of this action, new elements have appeared in view: ['tag': 'UL', 'content': 'No results found :(',, 'tag': 'a', 'content': 'Use full text search instead']. This means that the action of entering text fake news detection is not yet executed and needs further interaction. Get all_fields DOM to complete the interaction.,*

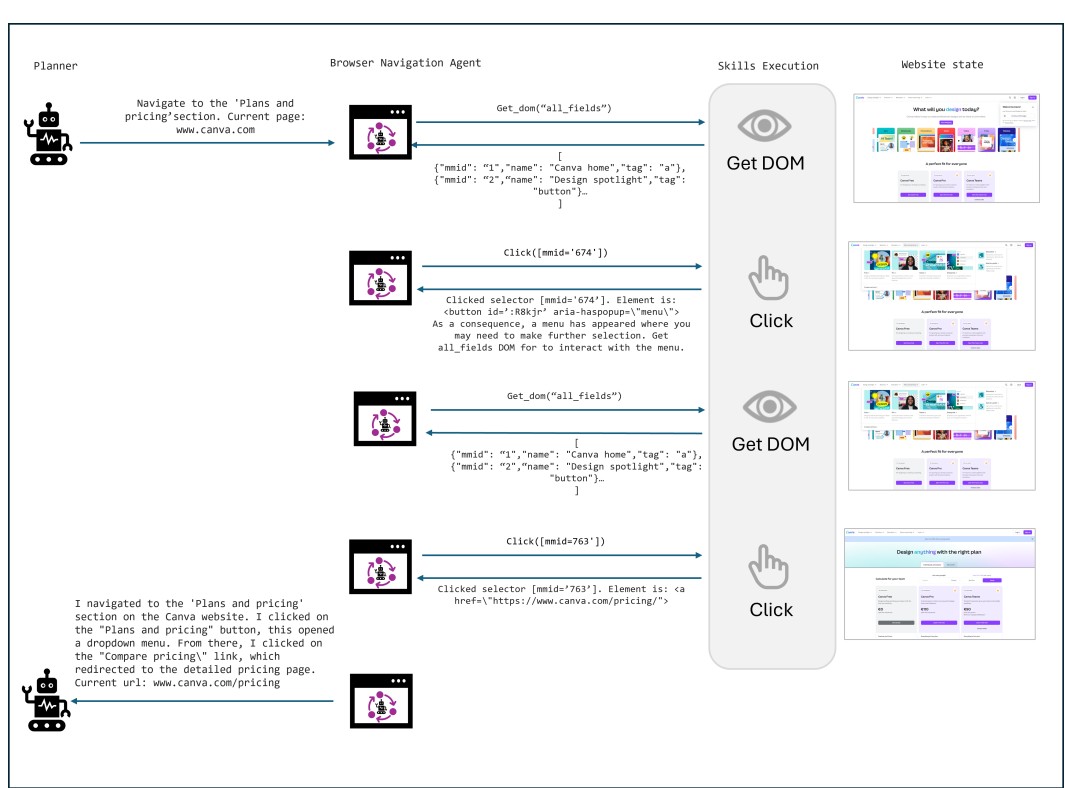

Figure 4: An example of Agent-E nested chat execution loop for the sub task *"Navigate to the plans and pricing section"* which is part of the larger task introduced earlier *"Find the price of Canva Teams subscription and minimum number of users required for it"*. The figure shows an example of change observation feedback as a result of change in aria-expanded attribute.

## F Agent-E Workflow Illustrations

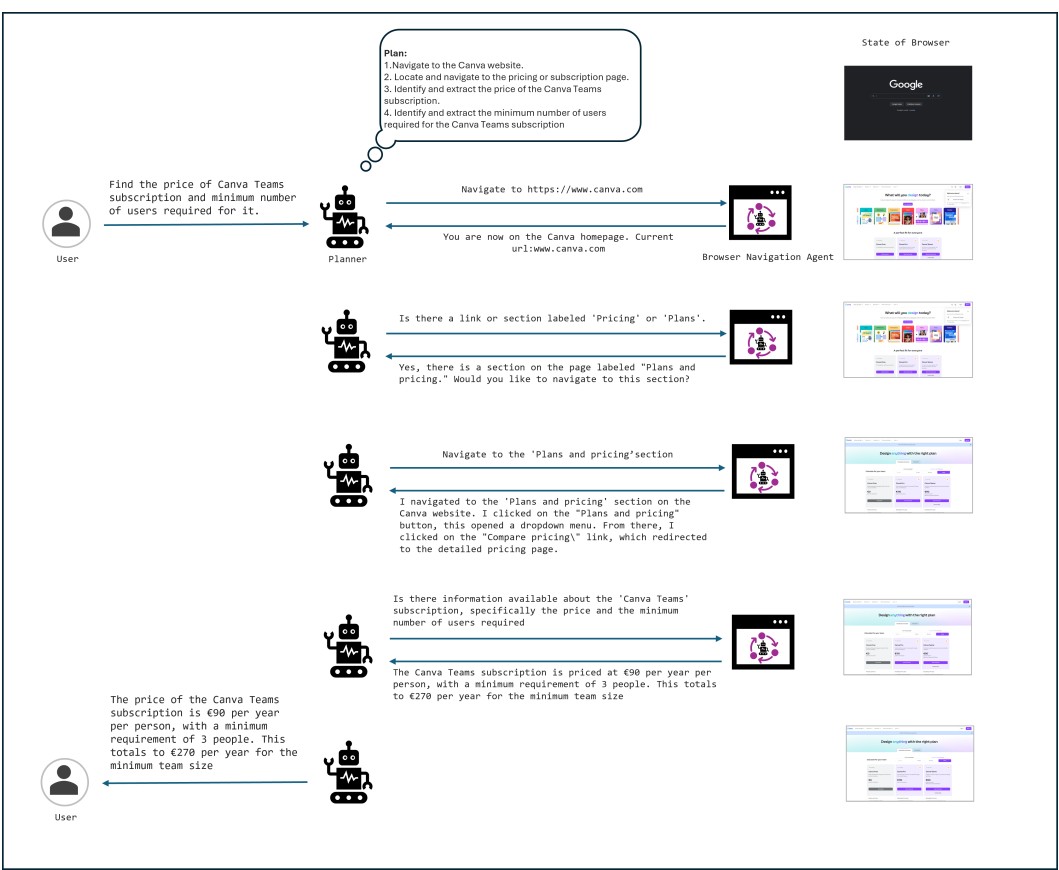

Figure 5: An example of Agent-E execution highlighting communication between the planner and browser navigation agent for the user task *Find the price of Canva Teams subscription and minimum number of users required for it*

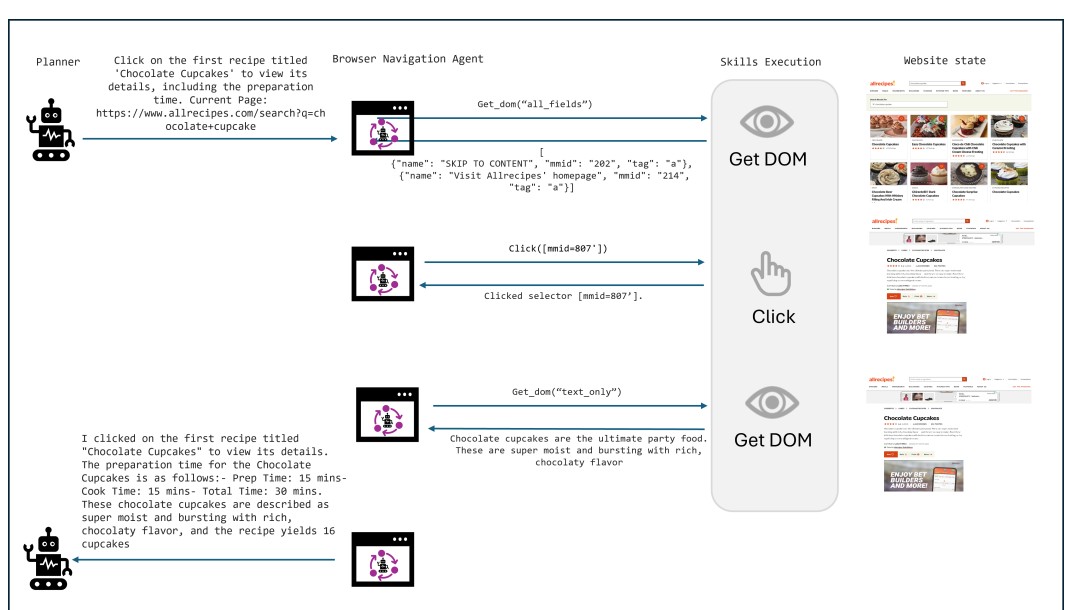

Figure 6: Providing multiple options for DOM observation allows to flexibly select one fit for task. The conversation is truncated with '...' to enhance readability in the image.

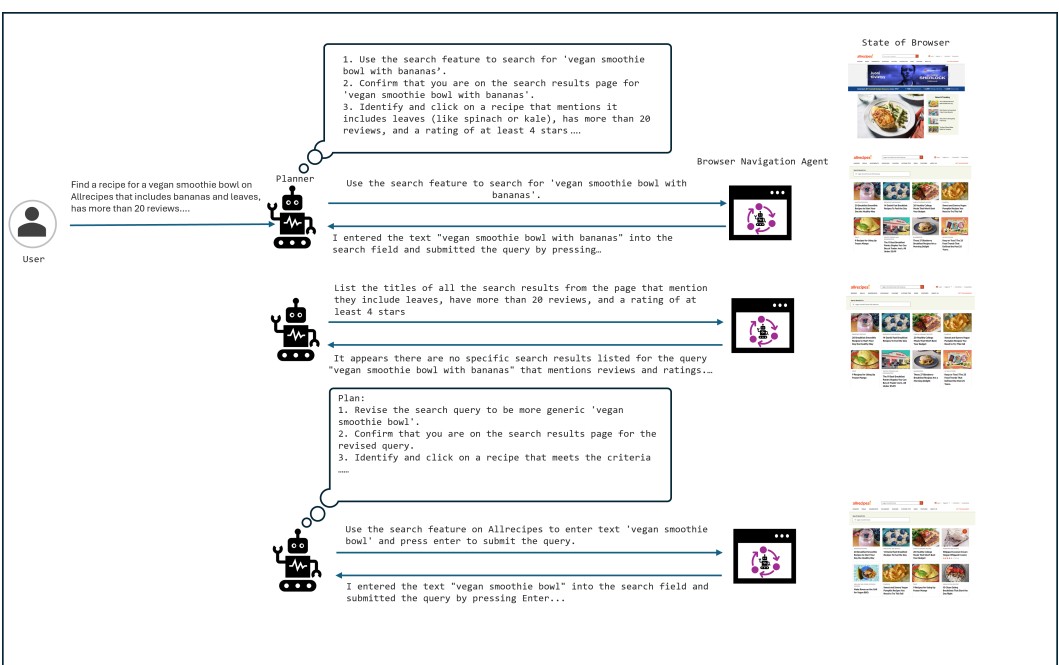

Figure 7: An example instance of Agent-E detecting and recovering from errors. The conversation is truncated with '...' to enhance readability in the image.

