# OpenReview forum: "Agent-E: From Autonomous Web Navigation to Foundational Design Principles in Agentic Systems"
_ICLR.cc/2025/Conference — Submitted to ICLR 2025_

### Official Review · Reviewer_UT8D · 2024-10-20

**Soundness:** 2
**Presentation:** 3
**Contribution:** 3
**Rating:** 6
**Confidence:** 3

**Summary:**

This paper presents Agent-E, an LLM-driven web agent designed to perform a range of web tasks including: page interaction, form filling, content summarisation, and analysis of DOM structures. Agent-E uses 3 LLM-powered agents to respectively perform high-level task planning, browser navigation to complete given tasks, and validation - in particular providing feedback on browser state when tasks are incomplete; allowing the agent to re-attempt the task and self-correct.

Further, the authors introduce 3 novel DOM Distillation strategies to pre-process the DOM that is presented to the LLM-powered agents. These are (1) text only - used in summarisation tasks (2) input fields - used in search or form-filling type interactions and (3) all fields - a complete JSON representation of all elements in the DOM. Additionally the authors provide change observations, such noting that popups appear when an LLM interacts with a button, to support the browser navigation agent in planning its next step.

**Strengths:**

- The paper tells a clear narrative, and does a good job of presenting the high level agent architecture, and capabilities of the agent.
 - Achieves SOTA performance on the WebVoyager benchmark, and justify use of this benchmark due to the diversity of pages available. The paper could be further strengthened by running their agent on the other benchmarks discussed in their paper.

**Originality**
The authors introduce several seemingly novel techniques that allow them to achieve SOTA performance on WebVoyager , including:
 - Use of distinct agents for high-level planning, browser navigation, and validation of success
 - Use of feedback from validation agent to re-attempt failed tasks
 - Use of DOM Distillation
 - Providing change actions to the browser navigation agent

**Quality**
The evaluation is thorough and displays the performance of different variations of the validation / refinement architecture across different websites in the benchmark.

**Clarity**
The paper clearly describes the high-level architecture of the agent, novel contributions and evaluation. However, it lacks various details that make it understand, e.g., the implementation of each agent (prompting and inputs) and does not have supplementary materials such as a codebase to facilitate this understanding.

**Significance**
The system achieves SOTA performance on the WebVoyager benchmark, beating previous models by over 16%.

**Weaknesses:**

- The authors choose to not make their code available for review. This makes it difficult to assess the accuracy with which the paper describes their codebase. Please provide anonymised repo using something like https://anonymous.4open.science/, and describe more details of your agent architecture in the appendix (i.e. prompts used for each agent). Morever, this limits the *theoretical* contributions of the paper, as various contributions of the work, are not described in great detail in the work. Such contributions include:
    - Change observation: No explanation is given of what information is given to the LLM to generate the natural language change observation; is it the DOM before and after? a diff? or some more novel algorithm that is applied?
     - What is the architecture of the validation agent / what information is it given to identify whether a task has been completed or not and give feedback
 - There are several claims that are not well quantified by the authors, including:
	 - "We consider the primitive skills we enabled in Agent-E to be enough for the vast majority of general web automation tasks": Perhaps there are statistics you can provide such as the number of tasks in the WebVoyager benchmark which require skills that are not enabled; and elaborate more on why
- The Agent Design Principles are based soley on the authors learnings and intuition, this section could be improved by drawing upon and referencing existing works that discuss architectures / design principles for (1) agentic software (2) LLM planning (3) LLM accuracy optimisation esp. when dealing with structured data. We also comment on some specific design principles:
	- "Routinely analyze, reflect": please use more precise language than "reflect"; it seems like you have (1) batch jobs that find common tasks and turn them into reproducible workflows that can be called (2) allow for tasks to be re-run with knowledge of outcomes from past tasks - much of this seems like optimisations for production settings, but not something that is particularly insightful from a scientific standpoint. I would have expected the word reflect to likely indicate fine tuning but that does not appear to be the case here.

**Questions:**

**Question**
 - Nitpick: Why did you choose the name verification agent, this confused me on the first read of the paper as I thought this agent would verify the *plan*, instead it seems that this agent is used to assess whether a task has succeeded after execution, and prompt re-attempt on failures. Perhaps something along the lines of "reviewer", "monitor" or "feedback" agent may be better.
 - "Hierarchical architecture excels in scenarios where tasks can be decomposed into sub-tasks that need to be handled at different levels of granularity"; realistically this just seems to be helping an LLM with Chain of Thought by getting it to decompose tasks at different levels of granularity giving it more time to "think". Have you run experiments to see if this hierarchical architecture still provides benefit when using models like o1-preview that are able to do this kind of chain-of-thought work out of the box.
 - Is DOM Distillation a term that the authors of this paper have coined, or is it used elsewhere?
 - What methodology, if any, was used to identify the 3 agent architecture - were there any other architectures that were tried before this?
 - Why was only the validation agent tested with vision modalities?

**Details Of Ethics Concerns:**

This paper presents an architecture for automated agents that can interact with websites to perform a task described in natural language. This can facilitate the development of a wide range of bots of potentially malicious nature (e.g. spam bots).

We would encourage the authors to include an Ethics Statement discussing these implications.

---

> ### Comment · Reviewer_UT8D · 2024-11-19
> **Clarifications to official review**
>
> Following valuable feedback from the Associate Program Chairs, we provide the following clarifications:
>
> > No explanation is given of what information is given to the LLM to generate the natural language change observation; is it the DOM before and after? a diff? or some more novel algorithm that is applied?
>
> As an actionable step, please provide a step-by-step explanation of how the change observation is generated, including what inputs are used, how changes are detected, and how this information is formatted for the LLM.
>
> > Perhaps there are statistics you can provide such as the number of tasks in the WebVoyager benchmark which require skills that are not enabled; and elaborate more on why
>
> Further, could you discuss any limitations you encountered due to the current set of primitive skills, and is there any data that you have on what percentage of real-world web tasks their primitive skills can handle.
>
> > Agent Design Principles
>
> Further to enriching your design principles, you could also compare the design principles suggested with specific existing frameworks or principles in the literature on agentic software and LLM-based systems.

---

> ### Author Response · Authors · 2024-11-21
> **Response to W1 by Reviewer UT8D**
>
> *We thank the reviewer for their valuable feedback and constructive suggestions, which have helped us improve the clarity and rigor of our work. Your feedback has helped us refine our explanations and analyses, and we have addressed each of your comments in the responses below:*
>
> > The authors choose to not make their code available for review. This makes it difficult to assess the accuracy with which the paper describes their codebase.
>
> Thank you for making us aware of the method for anonymizing GitHub codebase. We intended to include the links to the GitHub repo in the paper after the review stage. Below is the anonymized repo for this paper.
> * Agent-E: https://anonymous.4open.science/r/Agent-E-7E43/README.md
> * Agent-E w/o Self-Refinement: https://anonymous.4open.science/status/Agent-E-17AE
>
> > What is the architecture of the validation agent / what information is it given to identify whether a task has been completed or not and give feedback?
>
> The validation agent is a prompted gpt-4o or gpt-turbo-preview model which is prompted to judge a given workflow. The prompts for the validation agents can be found in [ae/core/prompts.py](https://anonymous.4open.science/r/Agent-E-7E43/test/validation_agent/prompts.py). In our paper, we have three different validation agents, each using a different method to represent the workflow:
>
> * **Task Log (Text)**: The text-based implementation uses the chat log of interactions between the planner agent and a proxy agent. The proxy agent summarizes the actions taken by the low-level browser navigation agent. The chat between the two agents is provided in a JSON format. An example of this chat log can be found here: [chat_log_example.json](https://anonymous.4open.science/r/Agent-E-7E43/test/example_workflow/logs/logs_for_task_397/execution_logs_397.json).
> * **Screenshots (Vision)**: The vision-based implementation relies on a sequence of screenshots captured before and after each action during the workflow. These screenshots provide a visual trail of the agent's execution process, allowing the validation agent to analyze the changes in the environment and determine whether the task has been completed successfully.
> * **Hybrid Validation**: The hybrid method combines the inputs of the previous two approaches. It utilizes the sequence of screenshots (as in the vision-based approach) along with the final response from the planner agent. An example of a task and final response is provided below:
>
> ```
> Task (397): "Execute the user task \"If I start using Copilot Individual, how much US dollars will it cost per year and what features does it have?\" Current Page: https://github.com/"
>
> Final response: “The annual cost for Copilot Individual on GitHub is $100 USD. Features include:\\n\\n- **Chat**: Unlimited messages, context-aware coding support, debugging, and security assistance.\\n- **Code Completion**: Real-time suggestions and comments.\\n- **Smart Actions**: Inline chat, prompt suggestions, slash commands, context variables, and commit message generation.\\n- **Supported Environments**: IDE, CLI, and GitHub Mobile.\\n- **Management and Policies**: Public code filter.\\n\\nThis plan is ideal for individual developers, freelancers, students, and educators. It offers a free trial, and is also free for verified students, teachers, and maintainers of popular open source projects”
> ```
>
> For more details, our GitHub repository contains the [implementation](https://anonymous.4open.science/r/Agent-E-7E43/test/validation_agent/validator.py) of the validation agent.

---

> > ### Comment · Reviewer_UT8D · 2024-11-22
> >
> > Thankyou for including the anonymised code which includes demos - this is useful supplimentary material to have.
> >
> > Could you please explain how you agents have memory (e.g. to form fill in https://www.youtube.com/embed/B5PWBNBbmQU)?

---

> > > ### Author Response · Authors · 2024-11-26
> > > **Regarding memory shown in the video for form filling**
> > >
> > > The version of Agent-E that was evaluated and reported in the paper did not have any notion of long term memory.
> > > The OSS version had a simple static file (located at "\ae\user_preferences\user_preferences.txt") where a user of Agent-E could add any information (details for form filling, or preferences such as 'for shopping, i prefer to use Amazon') which could be useful for customising the system and enable usecases like form-filling that you saw in the demo video. The information in this file was simply appended to the context of the planner agent.
> > >
> > > This capability was turned off for the purpose of our evaluation, which is also why we do not discuss about this in the paper. However, readme of the Github repo had some of this information.

---

> ### Author Response · Authors · 2024-11-21
> **Response to W1 (continued) by Reviewer UT8D**
>
> > Change observation: No explanation is given of what information is given to the LLM to generate the natural language change observation
>
> Thank you for pointing out the need for a more detailed explanation of the implementation and definition of Change Observation. Below, we clarify how Change Observation works and the information it provides to the LLM and will add this content to the paper:
>
> Change Observation allows viewing changes in the DOM immediately following an action execution. Without the change observation feedback, we noticed that LLM would perform an action and assume that it was done correctly. The purpose of the change observation was to nudge the LLM if heuristically we believe further action may be required to complete the step.
> Identifying what has changed on a website as a consequence of an action is a non-trivial problem because websites are implemented using diverse approaches. For example, some websites dynamically add new elements to the Document Object Model (DOM) after an action (e.g., the auto-suggestions that appear when entering text in search bars like Google or Amazon). Other websites achieve similar effects by modifying properties like visibility, opacity, position or display styles of existing elements, without adding new ones. In Agent-E, we implement Change Observation using two complementary approaches:
>
> 1. **Tracking changes in [aria-expanded](https://developer.mozilla.org/en-US/docs/Web/Accessibility/ARIA/Attributes/aria-expanded) attributes**: The aria-expanded attribute is a standard accessibility feature that indicates whether a particular element (e.g., a menu or dropdown) is expanded or collapsed. By observing if aria-expanded changes from False to True, we can infer if the element has changed state (e.g. “Click action on the element [mmid=25] was performed successfully). As a consequence a menu has appeared where you may need to make further selections. Get all_fields DOM to complete the action.” a relatively straightforward approach that tells the LLM that a menu is now open and likely further actions are needed. This method works effectively on websites that adhere to accessibility standards, regardless of how the underlying site is implemented.
>
> The steps to viewing change observations using aria-expanded attributes are below:
>
> ```
> 1. LLM invokes an action skill (e.g. click on element with mmid 823)
> 2. We check if the element has an aria-expanded property and its value
> 3. Perform the click operation
> 4. Wait 100ms
> 5. We check the new aria-expanded property and if it toggled from False to True.
> 6. If no, return a standard response -- “Success. Executed JavaScript Click on element with selector: [mmid='823']
> 7. If yes, return an additional message -- “Success. Executed JavaScript Click on element with selector: [mmid='823']. As a consequence, a menu has appeared where you may need to make further selection. Get all_fields DOM to complete the action. ”
> ```
>
>
> 2. **Using a DOM [Mutation Observer](https://developer.mozilla.org/en-US/docs/Web/API/MutationObserver)**: Mutation observers are tools that monitor changes in the DOM, such as the addition or modification of elements. We use this mechanism to detect if new elements are added after an action. In our case, we listen to changes that relate to the addition of new elements (if developers of the website are using a different approach, e.g. toggling the visibility of existing elements, this will not return any changes). Before any action is invoked, we subscribe to a mutation observer on that page and listens to any changes during the skill execution and an additional 100ms.
>
> The mutation observer returns a list of new elements that were added and we return that list to the LLM with an additional message. The steps to viewing change observations using Mutation Observer attributes are below:
>
> ```
> 1. LLM invokes an action skill (e.g. enter text “fake news detection model” on element with mmid 122)
> 2. We subscribe to DOM mutation observer for the full page
> 3. Perform the enter text operation
> 4. Wait 100ms
> 5. Unsubscribe the DOM mutation observer
> 6. Analyse if any new elements were added during this window. If No, simply return a success message:
>      “Success. Text \"fake news detection model\" set successfully in the element with selector [mmid='122']
>
> 7. If new elements were added, return a short list of elements with the return message. In the above example, it would return:
>         "Success. Text \"fake news detection model\" set successfully in the element with selector [mmid='122'].\n As a consequence of this action, new elements have appeared in view: [{'tag': 'UL', 'content': 'No results found :('},, {'tag': 'a', 'content': 'Use full text search instead'}]. This means that the action of entering text fake news detection is not yet executed and needs further interaction. Get all_fields DOM to complete the interaction.",

---

> ### Author Response · Authors · 2024-11-21
> **Response to W2 by Reviewer UT8D**
>
> > 2. "We consider the primitive skills we enabled in Agent-E to be enough for the vast majority of general web automation tasks": Perhaps there are statistics you can provide such as the number of tasks in the WebVoyager benchmark that require skills that are not enabled, and elaborate more on why.
>
> Although we consider most tasks possible with the primitive skills (or action space) of Agent-E, there are several cases where Agent-E would benefit from additional skills. These cases are not necessarily impossible to accomplish without additional skills but would make the task significantly easier to accomplish. Below, we have identified 11 cases where additional skills would be beneficial:
>
> 1. Amazon: Search for women's golf polos in m size, priced between 50 to 75 dollars, and save the lowest priced among results.
> 2. Amazon: Browse black strollers within $100 to $200 on Amazon. Then find one Among these black strollers with over 20,000 reviews and a rating greater than 4 star.
> 3. Amazon: Search for a wireless ergonomic keyboard with backlighting and a rating of at least 4 stars. The price should be between $40 to $60. Save the product with the 500+ customer reviews.
> 4. Amazon: Find a stainless steel, 12-cup programmable coffee maker on Amazon. The price range should be between $100 to $200. Report the one with the 4+ customer rating.
> 5. Amazon: Search for a queen-sized, hypoallergenic mattress topper on Amazon. It should have a memory foam material and be priced between $50 to $100.
> 6. Amazon: Find a compact digital camera on Amazon with a zoom capability of at least 10x, rated 4 stars or higher, and priced between $100 to $300.
> 7. Amazon: Find a portable Bluetooth speaker on Amazon with a water-resistant design, under $50. It should have a minimum battery life of 10 hours.
> 8. Booking: Find a hotel room on January 3-6 that is closest to National University of Singapore and costs less than $500
> 9. BBC: Find a AI-related story under Technology of Business. What is in the first picture in the story?
> 10.  BBC: Find a picture in the travel section that contains food, tell me what the food is called and what region it comes from.
> 11. Apple: Browse Apple Music on the entertainment section of the Apple's website, and see which singers' names are included in the pictures on this page.
>
> For Amazon and Booking.com examples, the ability to directly interact with price sliders would greatly streamline the process. While there are alternative methods to gather this information (e.g., sorting results by price and manually scrolling), these are more time-consuming, need more steps, and consequently, less efficient.
>
> The BBC tasks, on the other hand, appeared to require vision or image-understanding capabilities. However, BBC provides rich accessibility descriptions for most visual content, allowing these tasks to be completed using text-based methods alone. Similarly, while Apple’s website includes textual descriptions for some images, this coverage is incomplete, making full automation of the task infeasible using the current skill set of Agent-E.
>
> **Overall, these 11 tasks represent about 1.7% of the total 643 tasks and Agent-E performed 7 out of 11 of these tasks accurately.**

---

> ### Author Response · Authors · 2024-11-21
> **Response to W3 by Reviewer UT8D**
>
> > 3. The Agent Design Principles are based solely on the authors' learnings and intuition, this section could be improved by drawing upon and referencing existing works that discuss architectures/design principles for (1) agentic software (2) LLM planning (3) LLM accuracy optimization esp. when dealing with structured data. We also comment on some specific design principles.
>
> Given the growing body of work on LLM-based agents, we included the “Agent Design Principles” section to provide valuable insights for future practitioners. To address the reviewer's concern, we will make revisions to our Agentic Design Principles to include more precise language. To further address the reviewer’s concern about prior literature, we provide a summary of prior literature on 1) agentic software and 2) LLM planning and its connection to our work:
>
> 1. **Agentic Software Architecture**: Hewitt’s actor model [1] laid the groundwork for modern software agents by introducing self-contained, concurrent entities that interact through message-passing. This foundational concept directly informs our principle of **collaborative communication**, which emphasizes efficient, purpose-driven information exchange among agents. The actor model’s emphasis on modularity is also central to our principle of **adaptive modularity**, where agents are designed with distinct roles to support scalability and specialization. Franklin and Graesser’s taxonomy [2] builds on this by defining autonomous agents as systems situated in an environment that can sense, act, and adapt over time. This definition underpins our principle of **continuous learning and adaptability**, emphasizing agents’ ability to dynamically refine their behavior in response to new inputs. Their focus on goal-directed action also connects to reflect, and optimize on past experience, where agents analyze outcomes to improve future decision-making. Recent advancements in Multi-Agent Systems (MAS) [3] extend these foundational ideas, focusing on how groups of autonomous agents can collaborate to achieve complex goals. MAS frameworks often employ hierarchical architectures, where agents operate at varying levels of abstraction. This directly influences our principle to **adopt hierarchical architectures**, ensuring scalability and effective task delegation. Additionally, MAS research highlights the challenges of agent communication—too much information can overwhelm decision-making, while too little can reduce coordination. These insights are reflected in our principle of **collaborative communication**, which seeks to balance information sharing with task efficiency.
>
> 2. **LLM Planning**: Large Language Models (LLMs) have demonstrated remarkable capabilities in in-context learning and reasoning, but their application to real-world problems often requires grounding in specific tasks. Recent studies [4,5] explore hierarchical planning approaches combining LLMs with reinforcement learning (RL), enabling agents to perform both high-level reasoning and task-specific adaptation. These methods informed our principles of **continuous learning and adaptability** and **reflect, and optimize on past experiences**, where agents dynamically refine workflows and strategies based on iterative feedback. Other work, such as LLM-augmented Monte Carlo Tree Search (MCTS) [6,7], showcases the utility of LLMs in improving decision-making processes through exploration and iterative refinement. These advancements align with our iterative self-optimization principle, emphasizing the importance of post-task analysis to enhance agent performance. Furthermore, the taxonomy provided by Huang et al. [8] categorizes LLM-based planning methods into frameworks like REFLEXION and memory-augmented planning. These approaches validate our emphasis on creating agents capable of introspection and self-improvement.
>
> We have provided connections between our design principles and foundational concepts in agentic software and autonomous systems. These connections showcase connections to our design principles: modularity, adaptability, and reflection & optimization on experience, demonstrating the relevance of our principles to the broader research landscape.
>
> > "Routinely analyze, reflect": please use more precise language than "reflect"
>
> Our revision will include changing the language in the fifth design principle from “Routinely Analyze, Reflect, and Optimize Based on Past Experiences” to “Leverage Past Experience” to be more precise.

---

> > ### Author Response · Authors · 2024-11-21
> > **Citations to W3 by Reviewer UT8D**
> >
> > [1] Hewitt, C. (1977), “Viewing Control Structures as Patterns of Passing Messages”, Artificial Intelligence 8(3), 323-364.
> >
> > [2] Franklin, S. and Graesser, A. (1997) Is It an Agent, or Just a Program? A Taxonomy for Autonomous Agents, In: Müller, J.P., Wooldridge, M.J. and Jennings, N.R., Eds., Intelligent Agents III Agent Theories, Architectures, and Languages, Springer, Berlin Heidelberg, 21-35.
> >
> > [3] Masterman, T., Besen, S., Sawtell, M., & Chao, A. (2024). The Landscape of Emerging AI Agent Architectures for Reasoning, Planning, and Tool Calling: A Survey. ArXiv, abs/2404.11584.
> >
> > [4] Prakash, B., Oates, T., & Mohsenin, T. (2023). LLM Augmented Hierarchical Agents. ArXiv, abs/2311.05596.
> >
> > [5] Dalal, M., Chiruvolu, T., Chaplot, D.S., & Salakhutdinov, R. (2024). Plan-Seq-Learn: Language Model Guided RL for Solving Long Horizon Robotics Tasks. ArXiv, abs/2405.01534.
> >
> > [6] Zhou, A., Yan, K., Shlapentokh-Rothman, M., Wang, H., & Wang, Y. (2023). Language Agent Tree Search Unifies Reasoning Acting and Planning in Language Models. ArXiv, abs/2310.04406.
> >
> > [7] Putta, P., Mills, E., Garg, N., Motwani, S.R., Finn, C., Garg, D., & Rafailov, R. (2024). Agent Q: Advanced Reasoning and Learning for Autonomous AI Agents. ArXiv, abs/2408.07199.
> >
> > [8] Huang, X., Liu, W., Chen, X., Wang, X., Wang, H., Lian, D., Wang, Y., Tang, R., & Chen, E. (2024). Understanding the planning of LLM agents: A survey. ArXiv, abs/2402.02716.

---

> > > ### Author Response · Authors · 2024-11-21
> > > **Response to Q2, Q3, Q5 by Reviewer UT8D**
> > >
> > > > 2. "Hierarchical architecture excels in scenarios where tasks can be decomposed into sub-tasks that need to be handled at different levels of granularity"; realistically this just seems to be helping an LLM with Chain of Thought by getting it to decompose tasks at different levels of granularity giving it more time to "think". Have you run experiments to see if this hierarchical architecture still provides benefits when using models like o1-preview that can do this kind of chain-of-thought work out of the box?
> > >
> > > We appreciate the reviewer’s insightful comment on the potential overlap between hierarchical architectures and the native chain-of-thought (CoT) capabilities of modern models like o1-preview. While we have not yet run experiments with o1-preview or similar models, this is primarily due to the current limitations of the Autogen framework, which does not fully support these models. For instance, Autogen does not accommodate the “system” role or certain key parameters like temperature, which are essential for leveraging models such as o1-preview effectively. To isolate the impact of hierarchical architecture, we conducted experiments using GPT-4-Turbo with a single-agent system employing CoT-style prompting. This evaluation was performed using a subset of WebVoyager (75 tasks = 5 tasks randomly sampled tasks from each website * 15 websites). The results are presented below. (Note that the single agent system mentioned here makes use of other components of Agent-E such as change observation and DOM distillation.)
> > >
> > > |  |  |  |  |
> > > |---|---|---|---|
> > > |  | **Success Rate** | **Task Completion Time \(seconds\)** | **Avg\. LLM Calls** |
> > > | Single Agent System \(GPT\-4\-Turbo\) | 48% | 68\.2 | 9\.2 |
> > > | Hierarchical System \(GPT\-4\-Turbo\) | 70\.6% | 170 | 29 |
> > >
> > > The single-agent system, while computationally efficient, often failed to complete tasks requiring multi-step reasoning, exploration, or retries. Specifically, it exhibited the following limitations:
> > > 1. Premature Abandonment: Tasks were frequently left incomplete after initial failures.
> > > 2. Partial Completion: Responses often provided partial answers without finishing the full task.
> > > 3. Context Window Saturation: Accumulated noisy context information led to confusion and repeated navigation loops.
> > >
> > > In contrast, our hierarchical system achieved a 22.6% improvement in success rate by leveraging task decomposition and clear separation of responsibilities. This architecture excels in long-horizon workflows, enabling effective retries and backtracking when sub-tasks fail. We plan to add this experiment to our paper as additional supporting evidence for our hierarchical architecture.
> > >
> > > >  3. Is DOM Distillation a term that the authors of this paper have coined, or is it used elsewhere?
> > >
> > > In this paper, we introduce and coin the term "Flexible DOM Distillation". The need to filter semantically irrelevant content from HTML structures is a well-documented challenge in the web agent literature. Different works have addressed this under various terminologies, such as “HTML-Denoising” (as seen in A Real-World WebAgent with Planning, Long Context Understanding, and Program Synthesis) and “HTML Cleaning” (used in Steward: Natural Language Web Automation). Unlike prior work, Agent-E's technique supports multiple DOM observation strategies (i.e. all_fields, input_fields, and text_only) which adapt to the task at hand. To emphasize this key distinction, we coined the term “Flexible DOM Distillation.”
> > >
> > > > 5. Why was only the validation agent tested with vision modalities?
> > >
> > > We appreciate the reviewer bringing up this point about the use of multimodality in web navigation since this is an active area of research. While prior work [1,2] has demonstrated the potential benefits of incorporating vision or multimodal information into web agents, this paper shows that a DOM-based system can outperform vision-based models. The proposed architecture accomplishes this by proposing a set of novel approaches, including 1) the use of a hierarchical architecture,  2) the use of self-refinement, and 3) flexible DOM distillation. We recognize that expanding this approach to other components of our system could provide valuable insights and is indeed a promising direction for future work.
> > >
> > > [1] He, Hongliang, et al. "WebVoyager: Building an End-to-End Web Agent with Large Multimodal Models." arXiv preprint arXiv:2401.13919 (2024).
> > >
> > > [2] Lutz, Michael, et al. "WILBUR: Adaptive In-Context Learning for Robust and Accurate Web Agents." arXiv preprint arXiv:2404.05902 (2024).

---

> > > > ### Author Response · Authors · 2024-11-21
> > > > **Response to Q4 by Reviewer UT8D**
> > > >
> > > > > 4. What methodology, if any, was used to identify the 3 agent architecture - were there any other architectures that were tried before this?
> > > >
> > > > Before arriving at a three-agent system, we identified the limitations of current web-navigation systems.  in its ability to handle long-horizon tasks. To address these challenges, we introduced hierarchical planning, inspired by its proven efficacy in handling complex, multi-step goals through task decomposition [2, 3, 4]. This led to the development of a two-agent system, comprising: 1) A high-level planner responsible for task decomposition and 2) a low-level browser navigation agent tasked with executing subtasks. We present comparative results of this two-agent system against the single-agent approach on the WebVoyager dataset [1] in Table 1 of our paper.
> > > >
> > > > While the two-agent architecture improved performance overall, we observed that nearly half of the failures were self-aware, as detailed in Tables 4 and 5. These failures revealed the possibility of a self-correcting agent. Prior work has shown LLM-based iterative refinement and feedback systems [5, 6] to work well in other multi-step reasoning or planning settings, so we introduced a validation and feedback agent as a third component. The resulting three-agent architecture improved task performance, as evidenced in Table 2, where we compare it with the two-agent system. By adding each additional agent to our proposed system, we were able to show improvement in the overall abilities of the agent.
> > > >
> > > > [1] He, Hongliang, et al. "WebVoyager: Building an End-to-End Web Agent with Large Multimodal Models." arXiv preprint arXiv:2401.13919 (2024).
> > > >
> > > > [2] Wang, Z., Cai, S., Chen, G., Liu, A., Ma, X., and Liang, Y. (2022). Describe, explain, plan and select: Interactive planning with large language models enables open-world multitask agents. Advances in Neural Information Processing Systems, 37
> > > >
> > > > [3]  Nau, D., Cao, Y., Lotem, A., and Mu˜noz-Avila, H. (1991). Shop: Simple hierarchical ordered planner. International Joint Conference on Artificial Intelligence.
> > > >
> > > > [4] Marthi, B., Russell, S., and Wolfe, J. (2007). Angelic semantics for high-level actions. International Conference on Automated Planning and Scheduling.
> > > >
> > > > [5] Madaan, Aman, et al. "Self-refine: Iterative refinement with self-feedback." Advances in Neural Information Processing Systems 36 (2024).
> > > >
> > > > [6] Shinn, Noah, et al. "Reflexion: Language agents with verbal reinforcement learning." Advances in Neural Information Processing Systems 36 (2024).

---

> > ### Comment · Reviewer_UT8D · 2024-11-22
> >
> > Please ensure all promised revisions are made to the paper prior to the review deadline - I have updated my rating under the assumption that these changes will be made.

---

> ### Author Response · Authors · 2024-11-21
> **Addressing Ethical Concerns From Reviewer UT8D**
>
> We will include the following ethical statement in the paper to address the potential of malicious use cases for our work.
>
> ### Ethics Statement:
>
> As web agents like Agent-E move beyond research prototypes, they can raise important ethical concerns. First, web agents that operate on a personal device may introduce privacy issues for the user. These agents may have access to user sensitive information including passwords and financial data. Second, such agents, if used by a malicious user, could potentially be used for harmful purposes like sending spam and unauthorized web scraping. Thirdly, the widespread deployment of web agents could violate websites’  terms of service. While our research advances the technical capabilities of web agents, we recognize the critical importance of understanding failure modes and potential risks before real-world deployment. We acknowledge that benchmark performance alone is insufficient for ensuring safe deployment. Future work must establish robust security frameworks, access controls, and oversight mechanisms before web agents can be safely entrusted with user data and credentials. We emphasize that human oversight remains essential for deploying these systems responsibly

---

> ### Author Response · Authors · 2024-11-28
> **Paper updates for Reviewer UT8D**
>
> Thank you for your valuable feedback on our paper. As promised, we have made the following updates to address your concerns:
>
> - Improved definition of *change observation* in Section 2.
> - Added additional details of our *change observation* method in Appendix E.
> - Added anonymized repos of our agent implementation to the paper.
> - Added an Ethics statement.
> - Added our ablation comparing single-agent vs hierarchical-agent to Appendix C.
>
> We appreciate your critiques and believe these revisions add significant value and clarity for future readers.

---

### Official Review · Reviewer_WFFR · 2024-11-04

**Soundness:** 3
**Presentation:** 2
**Contribution:** 2
**Rating:** 5
**Confidence:** 4

**Summary:**

The paper proposes a novel architecture for solving Web tasks, comprising a multi agent system with a planner agent, browser navigation agent and validation agent. Next to the agent architecture, the authors propose a novel preprocessing/action formulation, where the agent gets access to a hand-made API. The latter enables to get the DOM tree in different representation or additional fine-grained filtered information.

The new agent system is evaluated on the WebVoyager benchmark, where it is compared with the provided baselines of the benchmark (using gpt4-turbo) itself plus a recent text only approach. The results show that the new agent system is on par or better for the different sub-tasks, where an additional improvement can be seen when activating the validation agent (using gpt4-o).

**Strengths:**

- Sensible research direction, as LLMs can be of great use in automatizing various Web tasks
- Superior empirical results on existing benchmark, which includes representative Web tasks
- Key learnings are extracted from the proposed method, including the implmeneted task-specific agent design

**Weaknesses:**

- Unclear if added value comes from DOM API or multi-agent system. At this point, it would be of value to have ablations or a proper baseline with only one LLM which uses the API.
- Unclear if choice of gpt4-o for the validation agent has an impact on the results.
- Related work does not concisely depict the delta to other works, but simply list other works.
- No usage of open-source models, which could additionally be fine-tuned

**Questions:**

- Have you empiricially evaluated the impact of using provided API and the agent design? The impact of the validation agent was evaluated separately, so one can extract the added value.
- Does the task-specific agent design might have limitations to Web tasks or would it generically work well for any browser-based Web task?
- What is the motivation and influence of using gpt4-o as validation agent and not sticking to gpt4-turbo? Would the results be less competitive with a gpt4-turbo validation agent?
- Have you performaned fine-tuning experiments with open-source models?

---

> ### Author Response · Authors · 2024-11-21
> **Response to Q1 From Reviewer WFFR**
>
> *We thank you for your valuable insights and suggestions. Your feedback has helped us refine our explanations and analyses, and we have addressed each of your comments in the responses below:*
>
> > 1. Have you empirically evaluated the impact of using the provided API and the agent design? The impact of the validation agent was evaluated separately, so one can extract the added value.
>
> There are two main design components that differ from prior web-navigation agents, which we propose outside of the validation agent: 1) the use of a hierarchical planner 2) the use of flexible DOM distillation. To demonstrate the impact of these two design components on the overall web-navigation agent, we have performed two ablation studies to tease out the benefits introduced by the hierarchical planner (by comparing it with a single agent system) and flexible dom distillation (by comparing our approach with a simpler approach of using Accessibility Tree (AxTree) directly, used in prior work such as [1]). Our evaluation suggests that both the hierarchical architecture and flexible DOM Distillation provide an overall increase of 22.5% and 16% respectively in terms of task success rate.
>
> **Hierarchical Planning:** We compared the hierarchical planner against a single-agent system. Both configurations utilized other components of Agent-E, such as change observation and DOM distillation. The analysis is on a subset of WebVoyager (75 tasks = 5 tasks randomly sampled from each website * 15 websites).
>
> |  |  |  |  |
> |---|---|---|---|
> |  | **Success Rate** | **Task Completion Time \(seconds\)** | **Avg\. LLM Calls** |
> | Single Agent System \(GPT\-4\-Turbo\) | 48% | 68\.2 | 9\.2 |
> | Hierarchical System \(GPT\-4\-Turbo\) | 70\.6% | 170 | 29 |
>
> Agent-E with the hierarchical planner improves the task success rate by 22.6%. However, it introduces increased computational overhead. The single-agent system, despite its lower computational cost, often struggles with tasks requiring multiple steps, exploration, or backtracking. Common failure modes include giving up prematurely if early attempts fail and providing incomplete answers without finishing the task in full. In contrast, the hierarchical system leverages its structured architecture to break down complex tasks into manageable sub-tasks, allowing the agents to handle long-horizon workflows more effectively. Although this results in higher computational costs due to the additional steps required, it enables the system to complete these workflows successfully.
>
> **Flexible DOM Distillation:** To evaluate flexible DOM distillation, we compared its performance against using the AXTree directly. Both configurations utilized the hierarchical planner and change observation. The analysis is on a subset of WebVoyager (75 tasks = 5 tasks randomly sampled from each website * 15 websites). The results are summarized below:
>
> |  |  |  |  |
> |---|---|---|---|
> |  | **Success Rate** | **Task Completion Time \(seconds\)** | **Avg\. LLM Calls** |
> | Flexible DOM distillation | 70\.6% | 170 | 29 |
> | AXTree only | 54\.6% | 161 | 37 |
>
> Flexible DOM distillation improved the success rate by 16%, showcasing its ability to better distill task-relevant information from complex DOMs. However, AXTree-based processing was marginally faster due to the additional steps required for DOM enrichment in our approach, which typically adds 1–2 seconds per call depending on webpage complexity.
> Our findings highlight that the hierarchical planner and flexible DOM distillation are crucial design components that contribute significantly to Agent-E's overall performance. While the hierarchical planner enables better task decomposition and management, flexible DOM distillation ensures robust handling of complex observation spaces. These enhancements jointly advance the state of web agents, albeit at some computational cost.
>
> We will add these new analyses to the paper.
>
> [1] He, Hongliang, et al. "WebVoyager: Building an End-to-End Web Agent with Large Multimodal Models." arXiv preprint arXiv:2401.13919 (2024).

---

> ### Author Response · Authors · 2024-11-21
> **Response to Q2, Q3, Q4 From Reviewer WFFR**
>
> > 2. Does the task-specific agent design might have limitations to Web tasks or would it generically work well for any browser-based Web task?
>
> We would like to clarify that each component of Agent-E is task-agnostic by design, and the results presented in the paper do not use any task- or website-specific configuration or customization. The system comprises three distinct agents, each playing a unique role in the workflow:
>
> * **Planner Agent**: Responsible for high-level task decomposition, it breaks down complex user instructions into manageable sub-tasks.
> * **Browser Navigation Planner**: Focused on executing these sub-tasks, it translates them into fine-grained web interactions specific to the current state of the browser.
> * **Validation Agent**: Ensures task completion by monitoring the workflow and providing feedback in cases of incomplete or failed tasks.
>
> Each of these agents is designed to be task-agnostic. This means that Agent-E is capable of handling various browser-based tasks without requiring website-specific customizations. We describe the possibility of using specialized web-agents in our Agent design Principle (number 6) by using task and website-specific prompting and skills. Such customization could further enhance performance and remains an avenue for future exploration; however, the system presented in this paper does not rely on such specializations.
>
> > 3. What is the motivation and influence of using gpt4-o as a validation agent and not sticking to gpt4-turbo? Would the results be less competitive with a gpt4-turbo validation agent?
>
> To demonstrate the impact of gpt-4-o vs. gpt-4-turbo on validation accuracy, we have run a comparative experiment:
>
> |                           | **True Positive** | **True Negative** | **False Positive** | **False Negative** | **Validator Accuracy** |
> |--------------------------------|-------------------|-------------------|--------------------|--------------------|------------------------|
> | Task Log (gpt-4-turbo-preview) | 66.56             | 17.68             | 7.40               | 8.36               | 84.24                  |
> | Task Log (gpt-4o)              | 70.51             | 14.51             | 9.20               | 5.77               | 85.02                  |
>
>
> As shown by the table above, gpt4-turbo-preview outperforms gpt4-o marginally in terms of accuracy. In practice, we found that using gpt4-o resulted in significantly faster execution times and was cheaper while showing comparable accuracy.
>
> >4. Have you performed fine-tuning experiments with open-source models?
>
> In this paper, our primary focus is to introduce a robust web navigation system, Agent-E, that addresses the challenges of web automation through novel architectural improvements and design principles. While fine-tuning with open-source models presents an exciting avenue for exploration, it falls outside the scope of this work. We instead concentrate on demonstrating the efficacy of the proposed system for web-navigation. Incorporating fine-tuning experiments with open-source models would be a valuable direction for future work.

---

> > ### Comment · Reviewer_WFFR · 2024-11-26
> >
> > Thank you for your additional insights. I believe they add value to the manuscript. I will adapt my score if I can see the clarifications and novel analyses in an updated version of the paper.

---

> > > ### Author Response · Authors · 2024-12-03
> > > **Follow-Up to Reviewer WFFR**
> > >
> > > Thank you for your insightful comments. We have carefully addressed and incorporated all your suggestions into the revised version. Please let us know if you have any further questions or concerns.

---

> ### Author Response · Authors · 2024-11-28
> **Paper updates for Reviewer WFFR**
>
> Thank you for your thoughtful feedback and suggestions. To address your concerns, we have made the following modifications to the original paper:
>
> - Rewrote the Related Work section to highlight the novelty in our paper better.
> - Added a comparison between using gpt-4o and gpt-4-turbo-preview for the validation agent in Appendix B.1.
> - Added an ablation comparing single-agent systems to hierarchical planning agents in Appendix C.
> - Added an ablations comparing the use of Flexible DOM Distillation vs accessibility tree only to represent the DOM in Appendix D.
>
> We are grateful for your critiques and believe these updates enhance the paper’s clarity and value for future readers.

---

### Official Review · Reviewer_hWZC · 2024-11-04

**Soundness:** 3
**Presentation:** 3
**Contribution:** 4
**Rating:** 6
**Confidence:** 5

**Summary:**

The paper introduces Agent-E, a web agent designed to perform complex web-based tasks more efficiently. Agent-E employs a novel hierarchical architecture comprising three LLM-powered components: a planner agent, a browser navigation agent, and a verification agent.

The planner agent is responsible for high-level task management, breaking down user instructions into a sequence of manageable subtasks. These are delegated to the browser navigation agent, which plans and executes the necessary lower-level actions to complete each subtask. To handle the complexity of DOMs and improve interpretability, the browser agent utilizes a flexible DOM distillation approach, selecting the most suitable DOM representation for each task to highlight key elements and avoid overwhelming the LLM with unnecessary information. Additionally, the agent employs a 'change observation' mechanism, inspired by the Reflexion paradigm, where it monitors state changes after each action and receives verbal feedback to enhance situational awareness and performance.
Agent-E also incorporates a verification agent that provides feedback on incomplete or failed tasks, enabling a self-correcting system through a self-refinement mechanism. Agent-E was tested in on the WebVoyager benchmark.

**Strengths:**

Originality
The paper demonstrates a nice degree of originality, primarily through the architectural approach in Agent-E. By introducing a hierarchical framework with distinct, specialized roles (planner agent, browser navigation agent, and validation agent), the authors effectively address several challenges in web automation. The flexible DOM distillation approach is another contribution, as it allows the browser navigation agent to dynamically tailor DOM representations to the specific needs of each task. This feature moves beyond static DOM handling methods seen in prior work, reducing cognitive load and enhancing accuracy. Furthermore, the self-refinement mechanism, inspired by a Reflexion-like paradigm, adds a unique layer of adaptability, allowing the agent to detect and correct failures in real-time. Together, these components present a good advancement over traditional web agents.

Quality
The paper is supported by an experimentation and evaluation on the WebVoyager benchmark. The authors provide a detailed comparison with both text-only and multimodal web agents, showing improvements over existing methods.

Clarity
The paper is generally clear and well-organized, with each component of Agent-E’s architecture clearly described. The role and function of the planner agent, browser navigation agent, and validation agent are each explained in detail, providing readers with a solid understanding of how Agent-E manages complex tasks. The authors also do a nice job of explaining the novel DOM distillation and change observation mechanisms (assuming you are reading the appendix).

Significance
Agent-E’s contribution looks significant in the field of autonomous web navigation, overcoming limitations in current web agents—particularly around handling complex, multi-step web tasks and interpreting lengthy and dynamic DOMs. The hierarchical architecture and the adaptive DOM distillation approach are likely to inspire future research on modular and adaptable agent architectures. The self-refinement mechanism also has broader implications, showcasing a feasible pathway for self-correcting agents that can enhance reliability in real-world applications. Given the increasing integration of LLM-powered agents in business and personal automation, Agent-E’s success rate and improved reliability on the WebVoyager benchmark underline its potential impact in advancing practical applications in web-based automation.

**Weaknesses:**

1. My main concern is regarding the limited benchmarking scope. While the paper presents results on the WebVoyager benchmark, the reliance on a single (one would say old) benchmark limits Agent-E’s effectiveness. Given the paper's goal to establish Agent-E as a state-of-the-art web agent, it must be evaluated on additional benchmarks: WorkArena, WebArena, ST-WebAgentBench.
This is a major weakness as I am not sure if the results will be the same on the SOTA benchmarks. I must admit that it is very hard for me to judge this agent based on the WebVoyager benchmark solely.

2. Agent-E’s architecture, with separate planner, browser, and validation agents, potentially introduces increased complexity and computational overhead. The paper does not fully address how this architecture scales in terms of computation and memory requirements, particularly when applied to larger, real-world workflows. Including benchmarks of computational resources used by Agent-E compared to simpler, single-agent systems would provide valuable insights.

**Questions:**

1. Can you add an explanation of DOM distillation, with performance analysis under different conditions?
2. Can you provide an in-depth study on the self-refinement mechanism’s impact on various error types and discuss potential trade-offs?
3. Can you include computational efficiency metrics and discuss optimizations or scalability considerations?

---

> ### Author Response · Authors · 2024-11-21
> **Response to Weaknesses By Reviewer hWZC**
>
> *Thank you for your thoughtful and constructive feedback. We have carefully reviewed your comments and provided detailed responses below:*
>
> > 1. My main concern is regarding the limited benchmarking scope. While the paper presents results on the WebVoyager benchmark, the reliance on a single (one would say old) benchmark limits Agent-E’s effectiveness. Given the paper's goal to establish Agent-E as a state-of-the-art web agent, it must be evaluated on additional benchmarks: WorkArena, WebArena, and ST-WebAgentBench. This is a major weakness as I am not sure if the results will be the same on the SOTA benchmarks. I must admit that it is very hard for me to judge this agent based on the WebVoyager benchmark solely.
>
> We believe that our evaluation of the WebVoyager dataset is sufficient to show the performance of Agent-E on real-life web navigation tasks. We chose WebVoyager for this study because it tests agent performance across 15 real-world websites with diverse and dynamic UI characteristics over 643 tasks. These include rich UI interactions (e.g., Booking.com), long-text processing (e.g., Wikipedia), and multi-step planning and replanning for complex tasks (e.g., AllRecipe and Amazon). Such tasks align closely with the challenges our work aims to address, making WebVoyager a robust and realistic evaluation platform for web agents. While WebArena and similar benchmarks feature representative tasks, their sandboxed environments and simplified UI implementations do not capture the real-world variability and complexity inherent in web-based user interfaces. For example, the complex date selectors in Booking.com and Google Flights are a UI element where all web agents have reportedly struggled (e.g. [1] and [2]). WebArena does not involve any such complex UI elements. For this reason, we believe WebVoyager reflects the unpredictability of dynamic content and browser behaviors, which we believe is crucial for evaluating an agent’s robustness and more difficult than benchmarks with synthetic environments. With that said we'd be happy to add results on other benchmarks in the final paper.
>
> [1] He, Hongliang, et al. "WebVoyager: Building an End-to-End Web Agent with Large Multimodal Models." arXiv preprint arXiv:2401.13919 (2024).
>
> [2] Lutz, Michael, et al. "WILBUR: Adaptive In-Context Learning for Robust and Accurate Web Agents." arXiv preprint arXiv:2404.05902 (2024).
>
> > 2. Agent-E’s architecture, with separate planner, browser, and validation agents, potentially introduces increased complexity and computational overhead. The paper does not fully address how this architecture scales in terms of computation and memory requirements, particularly when applied to larger, real-world workflows. Including benchmarks of computational resources used by Agent-E compared to simpler, single-agent systems would provide valuable insights.
>
> We agree that understanding the trade-offs in complexity, task completion time, and success rates is critical for real-world applications. Our paper includes computational evaluations in Appendix A. To address the cost in comparison to a single-agent system, we have performed an evaluation using a subset of WebVoyager (75 tasks = 5 tasks randomly sampled tasks from each website * 15 websites). The results are presented below. (Note that the single agent system mentioned here makes use of other components of Agent-E such as change observation and DOM distillation.)
>
> |  |  |  |  |
> |---|---|---|---|
> |  | **Success Rate** | **Task Completion Time \(seconds\)** | **Avg\. LLM Calls** |
> | Single Agent System \(GPT\-4\-Turbo\) | 48% | 68\.2 | 9\.2 |
> | Hierarchical System \(GPT\-4\-Turbo\) | 70\.6% | 170 | 29 |
>
> While the hierarchical system introduces increased computational overhead, the single-agent system performs significantly worse in terms of task success rates. The single-agent system, despite its lower computational cost, often struggles with tasks requiring multiple steps, exploration, or backtracking. Common failure modes include giving up prematurely if early attempts fail and providing incomplete answers without finishing the task in full. In contrast, the hierarchical system leverages its structured architecture to break down complex tasks into manageable sub-tasks, allowing the agents to handle long-horizon workflows more effectively, and allowing backtracking when a sub-task fails. Although this results in higher computational costs due to the additional steps required, it enables the system to complete these workflows successfully.
>
> We will update the paper to include these new results, along with a detailed discussion of the single-agent system's common error modes, in the appendix and main text.

---

> ### Author Response · Authors · 2024-11-21
> **Response to Questions By Reviewer hWZC**
>
> *Again, we thank you for your thoughtful and constructive feedback. We have carefully reviewed your comments and provided detailed responses below:*
>
> > 1. Can you add an explanation of DOM distillation, with performance analysis under different conditions?
>
> **DOM distillation** refers to the process of simplifying and extracting relevant parts of the Document Object Model (DOM) of a webpage. Raw HTML DOMs can be extremely large and noisy (e.g., YouTube homepage ~800,000 tokens). Processing such large and noisy inputs directly can overwhelm the underlying LLM. Our DOM distillation method consists of three DOM observation techniques which can be selected by the Browser Navigation Agent depending on the task:
>
> * **all_fields**: This is the most comprehensive DOM representation, provided in JSON format. It starts with the Accessibility Tree (AXTree) of the webpage—a simplified version of the DOM that omits non-semantic elements like <div> tags used purely for styling. We enrich this view with additional details, such as the names of HTML tags and inner text content where necessary. This representation is useful for tasks requiring detailed interaction with page elements.
> * **input_fields_only**: This is a subset of all_fields where only input fields and interactive elements from the DOM are returned. This strips away all the non-interactive text elements and allows the agents to use a much more succinct version of the DOM for purely interaction purposes.
> * **text_only**: This is a plain text view of the current page (gathered by using body.innertext in javascript of the current page). This will not have DOM identifiers to interact with screen elements but will have full text visible on the page. This is best suited for summarizing page content or answering specific questions from the page (e.g. what is the price of iPhone 16 or Is this product Waterproof?). Answering such questions with all_fields is a lot more challenging since the information can be fragmented across multiple DOM fields and thereby multiple JSON nodes.
>
> Previous web agents have also identified the issue with the expansive nature of HTML DOM and typically used the Accessibility Tree of the webpage directly (e.g. [1]). Below, we compare the Agent-E system (w/o self-refinement) using Flexible DOM distillation versus using only the accessibility tree (AXTree) to test the benefit of our DOM Distillation method. This analysis is on a subset of WebVoyager (75 tasks = 5 tasks randomly sampled from each website * 15 websites).
>
> |  |  |  |  |
> |---|---|---|---|
> |  | **Success Rate** | **Task Completion Time \(seconds\)** | **Avg\. LLM Calls** |
> | Flexible DOM distillation | 70\.6% | 170 | 29 |
> | AXTree only | 54\.6% | 161 | 37 |
>
> Flexible DOM distillation significantly improves success rates (+16%) by tailoring observations to task-specific needs. Using AXTree directly is marginally faster since the AXTree enrichment steps we perform for all_fields and input_fields take some processing time (typically an additional 1-2 seconds per call depending on the complexity of the webpage). These findings emphasize the importance of adaptive DOM distillation in enhancing Agent-E's effectiveness across diverse web navigation tasks.
>
>
> [1] He, Hongliang, et al. "WebVoyager: Building an End-to-End Web Agent with Large Multimodal Models." arXiv preprint arXiv:2401.13919 (2024).
>
> > 2. Can you provide an in-depth study on the self-refinement mechanism’s impact on various error types and discuss potential trade-offs?
>
> We appreciate the reviewer’s question regarding the self-refinement mechanism’s impact on error types and trade-offs. We are currently conducting this analysis and will share preliminary results before the end of the rebuttal period.
>
> > 3. Can you include computational efficiency metrics and discuss optimizations or scalability considerations?
>
> Thank you for your question regarding computational efficiency metrics and scalability considerations. In our response to Weakness 2, we have provided a comparison of task completion time and average LLM calls across Agent-E’s single-agent, two-agen on a subset of WebVoyager tasks. An additional computational efficiency breakdown is provided in Appendix A. These results highlight the trade-offs between computational cost and task success rates, as well as the scenarios where a hierarchical system is most beneficial.
>
> To summarize, while the hierarchical system introduces higher computational costs, it achieves significantly higher task success rates, particularly for complex workflows requiring multi-step planning and backtracking. By contrast, the single-agent system demonstrates lower computational cost but struggles with long-horizon tasks.

---

> > ### Comment · Reviewer_hWZC · 2024-11-26
> >
> > I appreciate the additional experiments and clarifications you have provided in response to my concerns. The new computational efficiency metrics and the detailed explanation of your DOM distillation approach have enhanced my understanding of your hierarchical architecture. Based on these updates, I am raising my score accordingly.
> >
> > However, I maintain that the absence of evaluations on key WebAgent benchmarks like WebArena remains a significant weakness of the paper. While I acknowledge your reasons for choosing WebVoyager and its merits in capturing real-world web complexities, including results on widely recognized benchmarks would strengthen the generalizability and impact of your work. Evaluating Agent-E on these benchmarks would provide a more comprehensive assessment of its performance relative to existing state-of-the-art web agents.
> >
> > Overall, your paper contributes valuable insights to the field of autonomous web navigation. Addressing the benchmarking scope further would enhance the paper's significance and applicability.

---

> ### Author Response · Authors · 2024-11-27
> **Response to Reviewer hWZC Q2**
>
> > Can you provide an in-depth study on the self-refinement mechanism’s impact on various error types and discuss potential trade-offs?
>
> To better understand the impact of using self-refinement, we labeled the subset of tasks that originally failed without refinement and succeeded with refinement. We saw 52 tasks improved with the task log validator, 38 improved with the screenshot validator, and 39 improved with the screenshot + final response validator. The errors were categorized into the following types:
>
> 1.  **Poor navigation**: These are tasks that the agent struggled to complete due to a lack of knowledge on how to navigate a specific website.
> 	- Apple Example: Agent is asked to *“Find information on Apple website, and tell me the device weight of Apple Vision Pro and list 5 Built-in Apps it supports”* The agent can navigate to the main Apple Vision Pro site but fails to navigate to the “tech spec” page which includes the apple vision pro weight.
>
> 2.  **Missing skills**: Due to limitations in Agent-E’s ability to interact with certain dynamic UI elements (e.g. filter on Amazon) or not view elements on the page without vision, some tasks become significantly more difficult to accomplish. These are tasks that would become significantly easier to accomplish with better UI interactions.
> 	- Booking.com Example: The agent is asked to *“Locate a hotel in Melbourne offering free parking and free WiFi, for a stay from August 28 to September 4, 2024”*. The agent can set the date and location correctly but because the agent cannot interact with the free parking and free wifi filter. While looking at the search results, the agent fails to find a hotel with free parking and free wifi after viewing the first few results. Then the agent gives up assuming no such hotel exits.
>
> 3.  **Incomplete Answer**: These are cases where the agent navigates to all the correct sites and takes all correct actions but doesn’t generate an incorrect or partial response.
> 	-  Apple example: Agent is asked *“Identify the most recent paper related to 'graph neural networks' on ArXiv and determine the affiliation of the first author”*. The agent correctly searches ‘graph neural network’ and pulls up the correct article, but fails to identify the affiliation and author.
>
> 4.  **DOM Interpretability**: The agent fails to complete the task because it cannot understand or find a key piece of information on the website DOM.
> 	- Google Flights example: The agent is asked to *“Search for a one-way flight from Mumbai to Vancouver on August 28, 2024, filtering the results to show only 1-stop flights”*. The agent searches for flights from Mumbia to Vancouver on the correct date, but is not able to identify the flights with only one stop due to issues interpreting the website.
>
> 5.  **Hallucinated Answer**: These are cases where the agent blatantly makes up an incorrect answer.
>
> | | **Task Log**    | **Screenshot** | **Screenshot + Final Response** |
> |---------------------------|-----------------|----------------|----------------------------------|
> |  **Error Type**            | |||
> | Missing Skill             | 40.38%         | 47.37%         | 51.28%                          |
> | Poor Navigation           | 25.00%         | 23.68%         | 28.21%                          |
> | DOM Interpretability      | 13.46%         | 7.89%          | 5.13%                           |
> | Incomplete Answer         | 5.77%          | 7.89%          | 10.26%                          |
> | Hallucinated Answer       | 1.92%          | 5.26%          | 0.00%                           |
> | Other                     | 13.46%         | 7.89%          | 5.13%                           |
> | **Total Samples**         | **51**             | **38**             | **39**                               |
>
>
>
>
> As depicted in the examples above, in **poor navigation** failures, the agent gives up on the task early, assuming that it is not possible to accomplish. In reality, the failure was due to their lack of expertise on how to navigate a particular website. A similar situation is true for **missing skill** and **DOM intepretability** failures. In most of these cases, the task is possible but the agent needs to account for its lack of inherent limitations on a particular website. Certain websites inherently have more dynamic UI elements or complex DOM structures, making them difficult for the agent to interact with effectively.
>
> The self-refinement mechanism encourages the agent to reflect and retry tasks, which helps the agent overcome initial failure points. While self-refinement improves performance (e.g., poor navigation failures), it comes with trade-offs. The mechanism can increase completion time and increase the number of LLM calls by re-executing failed tasks with alternative strategies. However, we view this trade as necessary for the agent to learn and explore the website further to accomplish the given task.

---

### Meta-Review · Area_Chair_Wic1 · 2024-12-20

**Metareview:**

This paper introduces Agent-E, a hierarchical LLM-powered web agent with innovative mechanisms like flexible DOM distillation and self-refinement. While the authors demonstrate promising results on the WebVoyager benchmark, a key weakness lies in the limited evaluation scope. Despite the authors' arguments for WebVoyager's suitability and their willingness to incorporate additional benchmarks in the final version, the current lack of generalizability raises concerns.

Strengths:

The hierarchical architecture, flexible DOM distillation, and self-refinement mechanism carry some novel ingredients to the field of web agents (hWZC, WFFR, UT8D). Agent-E achieves state-of-the-art performance on the WebVoyager benchmark (hWZC, WFFR, UT8D). On the presentation wise, the paper is well-written and clearly presents the agent's architecture and mechanisms (hWZC, UT8D).

Weaknesses:

The evaluation solely relies on the WebVoyager benchmark, limiting the generalizability of the results and raising concerns about the agent's performance on other established benchmarks like WebArena (hWZC, WFFR, UT8D). This is a significant weakness that hinders a comprehensive assessment of Agent-E's capabilities compared to existing state-of-the-art web agents.

Key Discussion Points:

DOM Distillation: The authors provided a detailed explanation and performance analysis of their flexible DOM distillation approach in response to reviewer questions.

Self-Refinement: An in-depth study on the self-refinement mechanism's impact on various error types was conducted, addressing reviewer hWZC's concerns.

Ablation Studies: Ablations were performed to assess the individual contributions of the DOM API and the multi-agent system, as requested by reviewer WFFR.

Change Observation: Reviewer UT8D's request for clarification on the implementation of change observation was met with a detailed explanation.

Agent Design Principles: The authors strengthened the theoretical grounding of the paper by connecting their design principles to existing works in response to reviewer UT8D's feedback.

Although Agent-E demonstrates promising results on the WebVoyager benchmark and the authors made efforts to address some of the concerns raised by the reviewers, the limited evaluation scope remains significant concerns about the generalizability of the findings.  The lack of evaluation on other established benchmarks prevents a comprehensive assessment of Agent-E's capabilities and its comparison to existing state-of-the-art web agents.

**Additional Comments On Reviewer Discussion:**

See the above meta review.

---

### Decision · Program_Chairs · 2025-01-22

Reject